# Graph-enhanced Optimizers for Structure-aware Recommendation Embedding Evolution

**Cong Xu**[†]    **Jun Wang**[†]    **Jianyong Wang** [§]    **Wei Zhang**[†*]

[†]East China Normal University    [§]Tsinghua University

[†]{congxueric, wongjun, zhangwei.thu2011}@gmail.com    [§]jianyong@tsinghua.edu.cn

## Abstract

Embedding plays a key role in modern recommender systems because they are virtual representations of real-world entities and the foundation for subsequent decision-making models. In this paper, we propose a novel embedding update mechanism, Structure-aware Embedding Evolution (SEvo for short), to encourage related nodes to evolve similarly at each step. Unlike GNN (Graph Neural Network) that typically serves as an intermediate module, SEvo is able to directly inject graph structural information into embedding with minimal computational overhead during training. The convergence properties of SEvo along with its potential variants are theoretically analyzed to justify the validity of the designs. Moreover, SEvo can be seamlessly integrated into existing optimizers for state-of-the-art performance. Particularly SEvo-enhanced AdamW with moment estimate correction demonstrates consistent improvements across a spectrum of models and datasets, suggesting a novel technical route to effectively utilize graph structural information beyond explicit GNN modules. Our code is available at https://github.com/MTandHJ/SEvo.

## 1 Introduction

Surfing Internet leaves footprints such as clicks [15], browsing [5], and shopping histories [58]. For a modern recommender system [6, 12], the entities involved (*e.g.*, goods, movies) are typically embedded into a latent space based on these interaction data. As the embedding takes the most to construct and is the foundation to subsequent decision-making models, its modeling quality directly determines the final performance of the entire system. According to the homophily assumption [31, 59], it is natural to expect that related entities have closer representations in the latent space. Note that the similarity between two entities refers specifically to those extracted from interaction data [47] or prior knowledge [37]. For example, goods selected consecutively by the same user or movies of the same genre are often perceived as more relevant. Graph neural networks (GNNs) [2, 10, 14] are a widely adopted technique to exploit such structural information, in concert with a weighted adjacency matrix wherein each entry characterizes how closely two nodes are related. Rather than directly injecting structural information into embedding, GNN typically serves as an intermediate module in the recommender system. However, designing a versatile GNN module suitable for various recommendation scenarios is challenging. This is especially true for sequential recommendation [8, 51], which needs to take into account both structural and sequential information at the same time. Moreover, the post-processing fashion inevitably increases the overhead of training and inference, limiting the scalability for real-time recommendation.

In this work, we aim to directly inject graph structural information into embedding through a novel embedding update mechanism. Figure 1 (a) illustrates a normal embedding evolution process, in

---

[*]Corresponding author. This work was supported in part by National Natural Science Foundation of China (No. 92270119 and No. 62072182) and Shanghai Institute of Artificial Intelligence for Education.

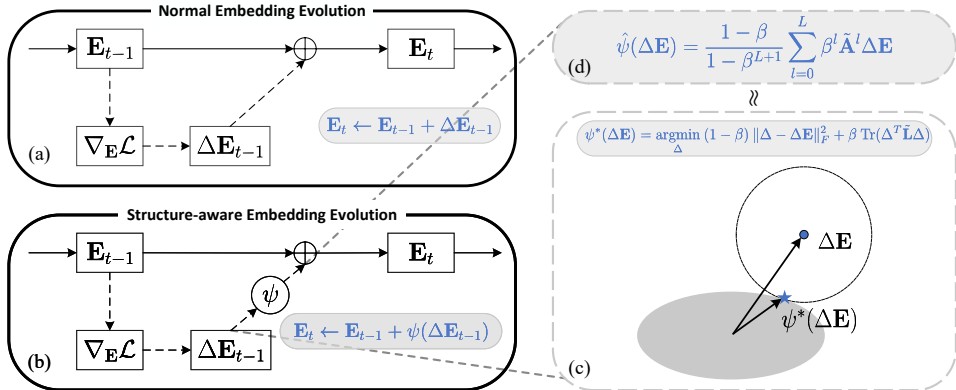

Figure 1: Overview of SEvo. (a) Normal embedding evolution. (b) (Section 2) Structure-aware embedding evolution. (c) (Section 2.2) Geometric visualization of the variation from $\Delta\mathbf{E}$ to $\psi^*(\Delta\mathbf{E})$. The gray ellipse represents the region with proper smoothness. (d) (Section 2.3) The $L$-layer approximation with a faster convergence guarantee.

which the embedding $\mathbf{E}$ is updated at step $t$ as follows:

$$\mathbf{E}_t \leftarrow \mathbf{E}_{t-1} + \Delta\mathbf{E}_{t-1}. \tag{1}$$

Note that the variation $\Delta\mathbf{E}$ is primarily determined by the (anti-)gradient. It points to a region able to decrease a loss function concerning recommendation performance [35], but lacks an explicit mechanism to ensure that the variations between related nodes are similar. The embeddings thus cannot be expected to capture model pairwise relations while minimizing the recommendation loss.

Conversely, structural information can be effectively learned if related nodes evolve similarly at each update. The structure-aware embedding evolution (SEvo), depicted in Figure 1 (b), is developed for this goal. A special transformation is applied so as to meet both smoothness and convergence [57]. Given that these two criteria inherently conflict to some extent, we resort to a graph regularization framework [57, 7] to balance them. While analogous frameworks have been used to understand feature smoothing and modify message passing mechanisms for GNNs [28, 60], applying this to variations is not as straightforward as to features [24] or labels [19]. Previous efforts are capable of smoothing, but cannot account for strict convergence. The specific transformation form must be chosen carefully; subtle differences may slow down or even kill training. Through comprehensive theoretical analysis, we develop an applicable solution with a provable convergence guarantee.

Apart from the anti-gradient, the variation $\Delta\mathbf{E}$ can also be derived from the moment estimates. Therefore, existing optimizers, such as SGD [41] and Adam [23], can benefit from SEvo readily. In contrast to Adam, AdamW [27] decouples the weight decay from the optimization step, making it more compatible with SEvo as it is unnecessary to smooth the weight decay as well. Furthermore, we recorrect the moment estimates of SEvo-enhanced AdamW when encountering sparse gradients. This modification enhances its robustness across a variety of recommendation models and scenarios. Extensive experiments over six public datasets have demonstrated that it can effectively inject structural information to boost recommendation performance. It is important to note that SEvo does not alter the inference logic of the model, so the inference time is exactly the same and very little computational overhead is required during training.

The main contributions of this paper can be summarized as follows. **1)** The graph regularization framework [57, 7] has been widely used for feature/label smoothing. To the best of our knowledge, we are the first to apply it to variations as an alternative to explicit GNN modules for recommendation. **2)** The final formulation of SEvo is non-trivial (previous iterative [57] or Neumann series [39] approximation methods proved to be incompatible in this case) and comes from comprehensive theoretical analyses. **3)** We further present SEvo-enhanced AdamW by integrating SEvo and recorrecting the original moment estimates. These modifications demonstrate consistent performance, yielding average 9%~23% improvements across a spectrum of models. For larger-scale datasets containing millions of nodes, the performance gains can be as high as 40%~139%. **4)** Beyond interaction

data, we preliminarily explore the pairwise similarity estimation based on other prior knowledge: node categories to promote intra-class representation proximity, and knowledge distillation [18] to encourage a light-weight student to mimic the embedding behaviors of a teacher model.

## 2 Structure-aware Embedding Evolution

In this section, we first introduce some necessary terminology and concepts, in particular smoothness. SEvo and its theoretical analyses will be detailed in Section 2.2 and 2.3. The proofs hereafter are deferred to Appendix A.

### 2.1 Preliminaries

**Notations and terminology.** Let $\mathcal{V} = \{v_1, \ldots, v_n\}$ denote a set of nodes and $\mathbf{A} = [w_{ij}] \in \mathbb{R}^{n \times n}$ a *symmetric* adjacency matrix, where each entry $w_{ij} = w_{ji}$ characterizes how closely $v_i$ and $v_j$ are related. They jointly constitute the graph $\mathcal{G} = (\mathcal{V}, \mathbf{A})$. For example, $\mathcal{V}$ could be a set of movies, and $w_{ij}$ is the frequency of $v_i$ and $v_j$ being liked by the same user. Denoted by $\mathbf{D} \in \mathbb{R}^{n \times n}$ the diagonal degree matrix of $\mathbf{A}$, the normalized adjacency matrix and the corresponding Laplacian matrix are defined as $\tilde{\mathbf{A}} = \mathbf{D}^{-1/2} \mathbf{A} \mathbf{D}^{-1/2}$ and $\tilde{\mathbf{L}} = \mathbf{I} - \tilde{\mathbf{A}}$, respectively. For ease of notation, we use $\langle \cdot, \cdot \rangle$ to denote the inner product, $\langle \mathbf{x}, \mathbf{y} \rangle = \mathbf{x}^T \mathbf{y}$ for vectors and $\langle \mathbf{X}, \mathbf{Y} \rangle = \text{Tr}(\mathbf{X}^T \mathbf{Y})$ for matrices. Here, $\text{Tr}(\cdot)$ denotes the trace of a given matrix.

**Smoothness.** Before delving into the details of SEvo, it is necessary to present a metric to measure the smoothness [57, 7] of node features $\mathbf{X}$ as a whole. Denoted by $\mathbf{x}_i, \mathbf{x}_j$ the row vectors of $\mathbf{X}$ and $d_i = \sum_j w_{ij}$, we have

$$\mathcal{J}_{smoothness}(\mathbf{X}; \mathcal{G}) := \text{Tr}(\mathbf{X}^T \tilde{\mathbf{L}} \mathbf{X}) = \sum_{i,j \in \mathcal{V}} w_{ij} \left\| \frac{\mathbf{x}_i}{\sqrt{d_i}} - \frac{\mathbf{x}_j}{\sqrt{d_j}} \right\|_2^2. \tag{2}$$

This term is often used as graph regularization for feature/label smoothing [24, 19]. A lower $\mathcal{J}_{smoothness}$ indicates smaller difference between closely related pairs of nodes, and in this case $\mathbf{X}$ is considered smoother. However, smoothness alone is not sufficient from a performance perspective. Over-emphasizing this indicator instead leads to the well-known over-smoothing issue [26]. How to balance variation smoothness and convergence is the main challenge to be addressed below.

### 2.2 Methodology

The entities involved in a recommender system are typically embedded into a latent space [6, 12], and the embeddings in $\mathbf{E} \in \mathbb{R}^{n \times d}$ are expected to be smooth so that related nodes are close to each other. As discussed above, $\mathbf{E}$ is learnable and updated at step $t$ by Eq. (1), where the variation $\Delta \mathbf{E}$ is mainly determined by the (anti-)gradient. For example, $\Delta \mathbf{E} = -\eta \nabla_{\mathbf{E}} \mathcal{L}$ when gradient descent with a learning rate of $\eta$ is used to minimize a loss function $\mathcal{L}$. However, the final embeddings based on this evolution process may be far from sufficient smoothness because: 1) The variation $\Delta \mathbf{E}$ points to the region able to decrease the loss function concerning recommendation performance, but lacks an explicit smoothness guarantee. 2) As numerous item embeddings (millions of nodes in practice) to be trained together for a recommender system, the variations of two related nodes may be quite different due to the randomness from initialization and mini-batch sampling.

We are to design a special transformation $\psi(\cdot)$ to smooth the variation so that the evolution deduced from the following update formula is structure-aware,

$$\mathbf{E}_t \leftarrow \mathbf{E}_{t-1} + \psi(\Delta \mathbf{E}_{t-1}). \tag{3}$$

Recall that, in this paper, the similarity is confined to quantifiable values in the adjacency matrix $\mathbf{A}$, in which more related pairs are weighted higher. Therefore, this transformation should encourage pairs of nodes connected with higher weights to evolve more similarly than those connected with lower weights. This can be boiled down to structure-aware transformation as defined below.

**Definition 1** (Structure-aware transformation). *The transformation $\psi(\cdot)$ is structure-aware if*

$$\mathcal{J}_{smoothness}(\psi(\Delta \mathbf{E})) \leq \mathcal{J}_{smoothness}(\Delta \mathbf{E}). \tag{4}$$

On the other hand, the transformation must ensure convergence throughout the evolution process, which means that the transformed variation should not differ too much from the original. For the sake of theoretical analysis, the ability to maintain the update direction will be used to *qualitatively* depict this desirable property below, though a quantitative squared error will be employed later.

**Definition 2** (Direction-aware transformation). *The transformation $\psi(\cdot)$ is direction-aware if*

$$\langle \psi(\Delta\mathbf{E}), \Delta\mathbf{E} \rangle > 0, \quad \forall \Delta\mathbf{E} \neq \mathbf{0}. \tag{5}$$

These two criteria inherently conflict to some extent. We resort to a hyperparameter $\beta \in [0, 1)$ to make a trade-off and the desired transformation is the corresponding minimum; that is,

$$\psi^*(\Delta\mathbf{E}; \beta) = \underset{\Delta}{\operatorname{argmin}} \, (1 - \beta) \, \|\Delta - \Delta\mathbf{E}\|_F^2 + \beta \, \operatorname{Tr}(\Delta^T \tilde{\mathbf{L}} \Delta). \tag{6}$$

A larger $\beta$ indicates a stronger smoothness constraint and $\psi^*(\Delta\mathbf{E})$ reduces to $\Delta\mathbf{E}$ when $\beta \to 0$. Geometrically, as shown in Figure 1 (c), $\psi^*(\Delta\mathbf{E})$ can be interpreted as a projection of $\Delta\mathbf{E}$ onto the region with proper smoothness. Taking the gradient to zero could give a closed-form solution, but it requires prohibitive arithmetic operations and memory overhead, which is particularly time-consuming in recommendation due to the large number of nodes. Zhou et al. [57] suggested a $L$-layer *iterative* approximation to circumvent this problem (with $\hat{\psi}_0(\Delta\mathbf{E}) = \Delta\mathbf{E}$):

$$\hat{\psi}_{iter}(\Delta\mathbf{E}) := \hat{\psi}_L(\Delta\mathbf{E}), \quad \hat{\psi}_l(\Delta\mathbf{E}) = \beta \tilde{\mathbf{A}} \hat{\psi}_{l-1}(\Delta\mathbf{E}) + (1 - \beta)\Delta\mathbf{E}.$$

The resulting transformation is essentially a momentum update that aggregates higher-order information layer by layer. Analogous message-passing mechanisms have been used in previous GNNs such as APPNP [24] and C&S [19]. However, this commonly used approximate solution is incompatible with SEvo; sometimes, variations after the transformation may be opposite to the original direction, resulting in a failure to converge.

**Theorem 1.** *The iterative approximation is direction-aware for all possible normalized adjacency matrices and $L \geq 0$, if and only if $\beta < 1/2$. In contrast, the Neumann series approximation $\hat{\psi}_{nsa}(\Delta\mathbf{E}) = (1 - \beta) \sum_{l=0}^{L} \beta^l \tilde{\mathbf{A}}^l \Delta\mathbf{E}$ is structure-aware and direction-aware for any $\beta \in [0, 1)$.*

As suggested in Theorem 1, a feasible compromise for $\hat{\psi}_{iter}$ is to restrict $\beta$ to $[0, 1/2)$, but this may cause a lack of smoothness. The Neumann series approximation [39] appears to be a viable alternative as it qualitatively satisfies both desirable properties. Nonetheless, this transformation can be further improved for a faster convergence rate based on the analysis presented next.

## 2.3 Convergence Analysis for Further Modification

In general, the recommender system has some additional parameters $\boldsymbol{\theta} \in \mathbb{R}^m$ except for embedding to be trained. Therefore, we analyze the convergence rate of the following gradient descent strategy:

$$\mathbf{E}_{t+1} \leftarrow \mathbf{E}_t - \eta \hat{\psi}_{nsa}\big(\nabla_{\mathbf{E}} \mathcal{L}(\mathbf{E}_t, \boldsymbol{\theta}_t)\big), \quad \boldsymbol{\theta}_{t+1} \leftarrow \boldsymbol{\theta}_t - \eta' \nabla_{\boldsymbol{\theta}} \mathcal{L}(\mathbf{E}_t, \boldsymbol{\theta}_t),$$

wherein SEvo is performed on the embedding and a normal gradient descent is applied to $\boldsymbol{\theta}$. To make the analysis feasible, some mild assumptions on the loss function should be given: $\mathcal{L} : \mathbb{R}^{n \times d} \times \mathbb{R}^m \to \mathbb{R}$ is a twice continuously differentiable function whose first derivative is Lipschitz continuous for some constant $C$. Then, we obtain the following properties.

**Theorem 2** (Informal). *If $\eta = \eta' = 1/C$, the convergence rate after $T$ updates is $\mathcal{O}(C/((1 - \beta)^2 T))$. If we adopt a modified learning rate of $\eta = \frac{1}{(1-\beta^{L+1})C}$ for embedding, the convergence rate could be improved to $\mathcal{O}(C/((1 - \beta)T))$.*

**Remark 1.** *Our main interest is to justify the designs of SEvo rather than to pursue a particular convergence rate, so some mild assumptions suggested in [32] are adopted here. By introducing the steepest descent for quadratic norms [1], better convergence can be obtained with stronger assumptions.*

Two conclusions can be drawn from Theorem 2. **1)** The theoretical upper bound becomes worse when $\beta \to 1$. This makes sense since $\hat{\psi}_{nsa}(\nabla_{\mathbf{E}} \mathcal{L})$ is getting smoother and further away from the

original descent direction. **2)** A modified learning rate for embedding can significantly improve the convergence rate. This phenomenon can be understood readily if we notice the fact that

$$\|\hat{\psi}_{nsa}(\Delta\mathbf{E})\|_F \leq (1 - \beta^{L+1})\|\Delta\mathbf{E}\|_F.$$

Thus, the modified learning rate is indeed to offset the *scaling effect* induced by SEvo. In view of this, we directly incorporate this factor into SEvo to avoid getting stuck in the learning rate search, yielding the final desired transformation:

$$\hat{\psi}(\Delta\mathbf{E}; \beta) = \frac{1 - \beta}{1 - \beta^{L+1}} \sum_{l=0}^{L} \beta^l \tilde{\mathbf{A}}^l \Delta\mathbf{E}. \tag{7}$$

It can be shown that $\hat{\psi}$ is structure-aware and direction-aware, and converges to $\psi$ as $L$ increases.

## 2.4 Integrating SEvo into Existing Optimizers

---

**Algorithm 1:** SEvo-enhanced AdamW. Differences from the original AdamW are colored in blue. The matrix operation below are element-wise.

---

**Input:** embedding matrix $\mathbf{E}$, learning rate $\eta$, momentum factors $\beta_1, \beta_2, \beta \in [0, 1)$, weight decay $\lambda$.

**foreach** *step $t$* **do**

$\quad\mathbf{G}_t \leftarrow \nabla_{\mathbf{E}}\mathcal{L}$ ;                                  `// Get gradients w.r.t E`

$\quad$Update first/second moment estimates for each node $i$:

$$\mathbf{M}_t[i] \leftarrow \begin{cases} \beta_1\mathbf{M}_{t-1}[i] + (1 - \beta_1)\mathbf{G}_t[i] & \text{if } \mathbf{G}_t[i] \neq \mathbf{0} \\ \beta_1\mathbf{M}_{t-1}[i] + \frac{(1-\beta_1)}{1-\beta_1^{t-1}}\mathbf{M}_{t-1}[i] & \text{otherwise} \end{cases},$$

$$\mathbf{V}_t[i] \leftarrow \begin{cases} \beta_2\mathbf{V}_{t-1}[i] + (1 - \beta_2)\mathbf{G}_t^2[i] & \text{if } \mathbf{G}_t[i] \neq \mathbf{0} \\ \beta_2\mathbf{V}_{t-1}[i] + \frac{(1-\beta_2)}{1-\beta_2^{t-1}}\mathbf{V}_{t-1}[i] & \text{otherwise} \end{cases};$$

$\quad$Compute bias-corrected first/second moment estimates:

$$\hat{\mathbf{M}}_t \leftarrow \mathbf{M}_t/(1 - \beta_1^t), \quad \hat{\mathbf{V}}_t \leftarrow \mathbf{V}_t/(1 - \beta_2^t);$$

$\quad$Update via SEvo:

$$\mathbf{E}_t \leftarrow \mathbf{E}_{t-1} - \eta\,\hat{\psi}\left(\hat{\mathbf{M}}_t/\sqrt{\hat{\mathbf{V}}_t + \epsilon}; \beta\right) - \eta\lambda\mathbf{E}_{t-1}.$$

**Output:** optimized embeddings $\mathbf{E}$.

---

SEvo can be seamlessly integrated into existing optimizers since the variation involved in Eq. (3) can be extended beyond the (anti-)gradient. For SGD with momentum, the variation becomes the first moment estimate, and for Adam this is jointly determined by the first/second moment estimates. AdamW is also widely adopted for training recommenders. Unlike Adam whose moment estimate is a mixture of gradient and weight decay, AdamW decouples the weight decay from the optimization step, which is preferable since it makes no sense to require the weight decay to be smooth as well. However, in very rare cases, SEvo-enhanced AdamW fails to work very well. We next try to ascertain the causes and then improve the robustness of SEvo-enhanced AdamW.

Denoted by $\mathbf{g} := \nabla_{\mathbf{e}}\mathcal{L} \in \mathbb{R}^d$ the gradient for a node embedding $\mathbf{e}$ and $\mathbf{g}^2 := \mathbf{g} \odot \mathbf{g}$ the element-wise square, AdamW estimates the first and second moments at step $t$ using the following formulas

$$\mathbf{m}_t = \beta_1\mathbf{m}_{t-1} + (1 - \beta_1)\mathbf{g}_{t-1}, \quad \mathbf{v}_t = \beta_2\mathbf{v}_{t-1} + (1 - \beta_2)\mathbf{g}_{t-1}^2,$$

where $\beta_1, \beta_2$ are two momentum factors. Then the original AdamW updates embeddings by

$$\mathbf{e}_t = \mathbf{e}_{t-1} - \eta \cdot \Delta\mathbf{e}_{t-1}, \quad \Delta\mathbf{e}_{t-1} := \hat{\mathbf{m}}_t/\sqrt{\hat{\mathbf{v}}_t}.$$

Note that the bias-corrected estimates $\hat{\mathbf{m}}_t = \mathbf{m}_t/(1 - \beta_1^t)$ and $\hat{\mathbf{v}}_t = \mathbf{v}_t/(1 - \beta_2^t)$ are employed for numerical stability [23]. In practice, only a fraction of nodes are sampled for training in a mini-batch, so the remaining embeddings have zero gradients. In this case, the sparse gradient problem may introduce some unexpected 'biases' as depicted below.

**Proposition 1.** *If a node is no longer sampled in subsequent $p$ batches after step $t$, we have* $\Delta\mathbf{e}_{t+p-1} = \kappa \cdot \frac{\beta_1^p}{\sqrt{\beta_2^p}}\Delta\mathbf{e}_{t-1}$, *and the coefficient of $\kappa$ is mainly determined by $t$.*

Considering a common case $\beta_2 \approx \beta_1$, the right-hand side approaches $\mathcal{O}(\beta_1^{p/2})$. The step size for inactive nodes then gets slower and slower during idle periods. This seems reasonable as their moment estimates are becoming outdated; however, this effect sometimes prevents the variation from being smoothed by SEvo. We hypothesize that this is because SEvo itself tends to assign more energy to active nodes and less energy to inactive nodes. So this auto-attenuation effect of the original AdamW is somewhat redundant. Fortunately, there is a feasible modification to make SEvo-enhanced AdamW more robust:

**Theorem 3.** *Under the same assumptions as in Proposition 1, $\Delta\mathbf{e}_{t+p-1} = \Delta\mathbf{e}_{t-1}$ if the moment estimates are updated in the following manner when $\mathbf{g}_t = \mathbf{0}$,*

$$\mathbf{m}_t = \beta_1\mathbf{m}_{t-1} + (1-\beta_1)\frac{1}{1-\beta_1^{t-1}}\mathbf{m}_{t-1}, \quad \mathbf{v}_t = \beta_2\mathbf{v}_{t-1} + (1-\beta_2)\frac{1}{1-\beta_2^{t-1}}\mathbf{v}_{t-1}. \quad (8)$$

As can be seen, when sparse gradients are encountered, the approach in Theorem 3 is actually to estimate the current gradient from previous moments. The coefficients $1/(1-\beta_1^{t-1})$ and $1/(1-\beta_1^{t-1})$ are used here for unbiasedness (refer to Appendix A.3 for detailed discussion and proofs). We summarize the SEvo-enhanced AdamW in Algorithm 1 and the modifications for Adam and SGD in Appendix B.1, with an empirical comparison presented in Section 3.3.

The previous discussion lays the technical basis for injecting graph structural information, but the final recommendation performance is determined by how 'accurate' the similarity estimation is. Following other GNN-based sequence models [48, 49], the number of consecutive occurrences across all sequences will be used as the pairwise similarity $w_{ij}$. In other words, items $v_i$ and $v_j$ that appear consecutively more frequently are assumed more related. Notably, we would like to emphasize that SEvo can readily inject other types of knowledge beyond interaction data. We have made some preliminary efforts in Appendix C and observed some promising results.

## 3 Experiments

In this section, we comprehensively verify the superiority of SEvo. We focus on sequential recommendation for two reasons: 1) This is the most common scenario in practice; 2) Utilizing both sequential and structural information is beneficial yet challenging. We showcase that SEvo is a promising way to achieve this goal. It is worth noting that although technically SEvo can be applied to general graph embedding learning [4], we contend SEvo-AdamW is especially useful for mitigating the inconsistent embedding evolution caused by data sparsity, while effectively injecting structural information in conjunction with other types of information.

Due to space constraints, this section presents only the primary results concerning accuracy, efficiency, and some empirical evidence that supports the aforementioned claims. We begin by introducing the datasets, evaluation metrics, baselines, and implementation details.

**Datasets.** Six benchmark datasets are considered in this paper. The first four datasets including Beauty, Toys, Tools, and MovieLens-1M are commonly employed in previous studies for empirical comparisons. Additionally, two larger-scale datasets, Clothing and Electronics, are used to assess SEvo's scalability in scenarios involving millions of nodes. Following [22, 13], we filter out users and items with less than 5 interactions, and the validation set and test set are split in a *leave-one-out* fashion,

Table 1: Dataset statistics

| Dataset | #Users | #Items | #Interactions | Avg. Len. |
|---|---|---|---|---|
| Beauty | 22,363 | 12,101 | 198,502 | 8.9 |
| Toys | 19,412 | 11,924 | 167,597 | 8.6 |
| Tools | 16,638 | 10,217 | 134,476 | 8.1 |
| MovieLens-1M | 6,040 | 3,416 | 999,611 | 165.5 |
| Electronics | 728,489 | 159,729 | 6,737,580 | 9.24 |
| Clothing | 1,219,337 | 376,378 | 11,282,445 | 9.25 |

namely the last interaction for testing and the penultimate one for validation. This splitting allows for fair comparisons, either for sequential recommendation or collaborative filtering. The dataset statistics are presented in Table 1.

**Evaluation metrics.** For each user, the predicted scores over *all items* will be sorted in descending order to generate top-N candidate lists. We consider two widely-used evaluation metrics, HR@N

Table 2: Overall performance comparison. The best baselines and ours are marked in underline and **bold**, respectively. Symbol ▲% stands for the relative gap between them. Paired t-test is performed over 5 independent runs for evaluating $p$-value ($\leq 0.05$ indicates statistical significance). 'Avg. Improv.' for each backbone depicts average relative improvements against the baseline.

| | | GNN-based | | | MF or RNN/Transformer-based | | | | | | | | | | ▲% | p-value |
|---|---|---|---|---|---|---|---|---|---|---|---|---|---|---|---|---|
| | LightGCN | SR-GNN | LESSR | MAERec | MF-BPR | +SEvo | GRU4Rec | +SEvo | SASRec | +SEvo | BERT4Rec | +SEvo | STOSA | +SEvo | | |
| **Beauty** HR@1 | 0.0074 | 0.0059 | 0.0088 | 0.0113 | 0.0071 | 0.0076 | 0.0061 | 0.0094 | 0.0120 | 0.0154 | 0.0157 | 0.0172 | 0.0166 | **0.0216** | 30.2% | 6.90E-05 |
| HR@5 | 0.0289 | 0.0247 | 0.0322 | 0.0424 | 0.0272 | 0.0293 | 0.0233 | 0.0326 | 0.0404 | 0.0499 | 0.0479 | 0.0522 | 0.0479 | **0.0544** | 13.5% | 7.69E-04 |
| HR@10 | 0.0472 | 0.0406 | 0.0506 | 0.0662 | 0.0454 | 0.0480 | 0.0395 | 0.0524 | 0.0634 | 0.0759 | 0.0716 | 0.0772 | 0.0680 | **0.0774** | 8.2% | 2.66E-03 |
| NDCG@5 | 0.0181 | 0.0152 | 0.0205 | 0.0269 | 0.0170 | 0.0184 | 0.0146 | 0.0209 | 0.0262 | 0.0328 | 0.0319 | 0.0350 | 0.0327 | **0.0383** | 17.2% | 9.70E-05 |
| NDCG@10 | 0.0240 | 0.0203 | 0.0264 | 0.0346 | 0.0228 | 0.0244 | 0.0198 | 0.0273 | 0.0336 | 0.0411 | 0.0395 | 0.0430 | 0.0391 | **0.0457** | 15.5% | 8.02E-05 |
| **Toys** HR@1 | 0.0087 | 0.0100 | 0.0126 | 0.0171 | 0.0079 | 0.0099 | 0.0059 | 0.0080 | 0.0172 | 0.0192 | 0.0160 | 0.0175 | 0.0232 | **0.0267** | 15.3% | 1.46E-03 |
| HR@5 | 0.0279 | 0.0294 | 0.0352 | 0.0532 | 0.0267 | 0.0306 | 0.0209 | 0.0276 | 0.0506 | 0.0584 | 0.0430 | 0.0492 | 0.0571 | **0.0625** | 9.6% | 5.70E-03 |
| HR@10 | 0.0456 | 0.0439 | 0.0513 | 0.0796 | 0.0427 | 0.0477 | 0.0345 | 0.0446 | 0.0727 | 0.0844 | 0.0645 | 0.0723 | 0.0776 | **0.0872** | 9.5% | 3.07E-04 |
| NDCG@5 | 0.0183 | 0.0198 | 0.0240 | 0.0355 | 0.0174 | 0.0203 | 0.0134 | 0.0179 | 0.0342 | 0.0392 | 0.0297 | 0.0336 | 0.0406 | **0.0453** | 11.6% | 2.48E-03 |
| NDCG@10 | 0.0240 | 0.0245 | 0.0292 | 0.0440 | 0.0225 | 0.0258 | 0.0177 | 0.0234 | 0.0413 | 0.0475 | 0.0366 | 0.0410 | 0.0472 | **0.0532** | 12.7% | 3.12E-05 |
| **Tools** HR@1 | 0.0067 | 0.0046 | 0.0045 | 0.0083 | 0.0058 | 0.0071 | 0.0053 | 0.0058 | 0.0099 | 0.0108 | 0.0074 | 0.0087 | 0.0095 | **0.0133** | 33.8% | 1.23E-02 |
| HR@5 | 0.0212 | 0.0162 | 0.0157 | 0.0271 | 0.0187 | 0.0225 | 0.0174 | 0.0208 | 0.0317 | 0.0337 | 0.0244 | 0.0279 | 0.0276 | **0.0350** | 10.5% | 8.94E-03 |
| HR@10 | 0.0326 | 0.0260 | 0.0263 | 0.0423 | 0.0293 | 0.0348 | 0.0272 | 0.0336 | 0.0466 | 0.0497 | 0.0405 | 0.0441 | 0.0417 | **0.0502** | 7.7% | 8.25E-03 |
| NDCG@5 | 0.0140 | 0.0103 | 0.0101 | 0.0177 | 0.0123 | 0.0147 | 0.0114 | 0.0133 | 0.0210 | 0.0223 | 0.0159 | 0.0183 | 0.0186 | **0.0244** | 15.9% | 5.57E-03 |
| NDCG@10 | 0.0176 | 0.0135 | 0.0135 | 0.0226 | 0.0157 | 0.0187 | 0.0145 | 0.0174 | 0.0258 | 0.0274 | 0.0211 | 0.0235 | 0.0231 | **0.0293** | 13.4% | 3.79E-03 |
| **MovieLens** HR@1 | 0.0124 | 0.0383 | 0.0513 | 0.0439 | 0.0117 | 0.0133 | 0.0487 | 0.0487 | 0.0490 | 0.0517 | 0.0681 | **0.0733** | 0.0457 | 0.0510 | 7.6% | 3.81E-02 |
| HR@5 | 0.0495 | 0.1297 | 0.1665 | 0.1563 | 0.0470 | 0.0509 | 0.1625 | 0.1663 | 0.1599 | 0.1670 | 0.2069 | **0.2127** | 0.1409 | 0.1569 | 2.8% | 4.17E-03 |
| HR@10 | 0.0866 | 0.2009 | 0.2539 | 0.2462 | 0.0836 | 0.0876 | 0.2522 | 0.2568 | 0.2492 | 0.2567 | 0.3018 | **0.3075** | 0.2185 | 0.2356 | 1.9% | 9.32E-02 |
| NDCG@5 | 0.0307 | 0.0842 | 0.1092 | 0.1003 | 0.0291 | 0.0319 | 0.1061 | 0.1075 | 0.1046 | 0.1096 | 0.1387 | **0.1437** | 0.0932 | 0.1041 | 3.6% | 4.19E-04 |
| NDCG@10 | 0.0427 | 0.1071 | 0.1373 | 0.1292 | 0.0408 | 0.0436 | 0.1350 | 0.1366 | 0.1333 | 0.1385 | 0.1693 | **0.1743** | 0.1181 | 0.1295 | 2.9% | 1.30E-02 |
| Avg. Improv. | | | | | | +13.1% | | +23.4% | | +12.3% | | +9.6% | | +17.5% | | |
| Avg. Train. Time | 2,820s | 43,783s | 62,457s | 31,674 | 2,863s | +173s | 3,582s | +124s | 532s | +37s | 1,256s | +288s | 2,087s | +127s | | |
| Avg. Inf. Time | 1.19s | 11.18s | 9.34s | 2.73s | 1.13s | +0s | 1.80s | +0s | 1.88s | +0s | 1.61s | +0s | 6.72 | +0s | | |

(Hit Rate) and NDCG@N (Normalized Discounted Cumulative Gain). The former measures the rate of successful hits among the top-N recommended candidates, while the latter takes into account the ranking positions and assigns higher scores in order of priority.

**Baselines.** We select four GNN-based models (LightGCN [16], SR-GNN [48], LESSR [8], and MAERec [51]) as performance and efficiency benchmarks. Since this study is not to develop a new model, four classic sequence models (GRU4Rec [17], SASRec [22], BERT4Rec [40], and STOSA [13]) are utilized as backbones to validate the effectiveness of SEvo. Besides, MF-BPR [35] is also considered here as a backbone without sequence modeling. We carefully tune the hyperparameters according to their open-source code and experimental settings.

**Implementation details.** Since the purpose is to study the effectiveness of SEvo, only hyperparameters concerning optimization are retuned, including learning rate ([1e-4, 5e-3]), weight decay ([0, 0.1]) and dropout rate ([0, 0.7]). Other hyperparameters in terms of the architecture are consistent with the corresponding baseline. For a fair comparison, the number of layers $L$ is fixed to 3 as in other GNN-based recommenders. As for the hyperparameter in terms of the degree of smoothness, we found $\beta = 0.99$ performs quite well in practice. The loss functions follow the suggestions in the respective papers, given that SEvo can be applied to any of them.

## 3.1 Overall Comparison

In this section, we are to verify the effectiveness of SEvo in boosting recommendation performance. Table 2 compares the overall performance and efficiency over four widely used datasets, and Table 3 further provides the results on two large-scale datasets with millions of nodes.

Firstly, GNN-based models seem to over-emphasize structural information but lack full exploitation of sequential information. Their performance is only on par with GRU4Rec. On the Tools dataset, SR-GNN and LESSR are even inferior to LightGCN, a collaborative filtering model with no access to sequential information. MAERec makes a slightly better attempt to combine the two by learning structural information through graph-based reconstruction tasks. It employs a SASRec backbone for recommendation to allow for a better utilization of sequential information. Despite the identical recommendation backbone, SASRec trained with SEvo-enhanced AdamW enjoys significantly better performance. Not to mention the consistent gains on other state-of-the-art methods such as BERT4Rec and STOSA, from 2% to 30% according to the last two columns of Table 2. Overall, the promising improvements from the SEvo enhancement suggest a different route to exploit graph structural information, especially in conjunction with sequence modeling.

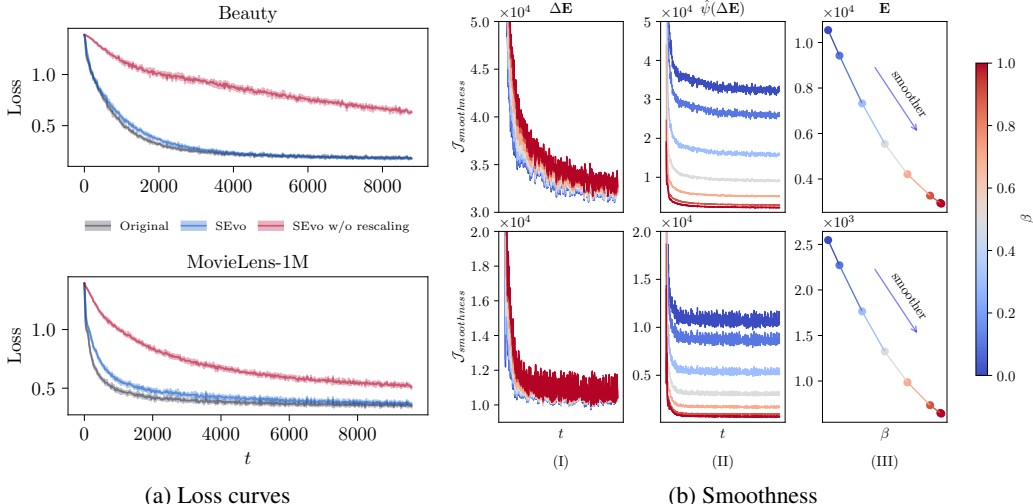

(a) Loss curves        (b) Smoothness

Figure 2: Empirical illustrations of convergence and smoothness. The top and bottom panels respectively depict the results for Beauty and MovieLens-1M. (a) SASRec enhanced by SEvo with or without rescaling. (b) Smoothness of (I) the original variation; (II) the smoothed variation; (III) the optimized embedding. A lower $\mathcal{J}_{smoothness}$ indicates stronger smoothness.

Secondly, either dynamic graph construction in SR-GNN and LESSR, or path sampling in MAERec, requires heavy computational overhead, which directly causes the training failures on large-scale datasets like Electronics and Clothing. Even worse, these high costs associated with SR-GNN and LESSR are inevitable during inference. In contrast, SEvo does not alter the model inference logic at all, thereby maintaining consistent inference time. The computational overhead required in training is also negligible compared to previous graph-enhanced models that employ GNNs as intermediate modules. For example, SASRec with SEvo consumes only 10 minutes compared to the hours of training time required for MAERec. When millions of nodes are encountered in Table 3, each epoch

Table 3: SEvo on large-scale datasets.

| | | SASRec | +SEvo | ▲% |
|---|---|---|---|---|
| Electronics | HR@1 | 0.0033 | **0.0063** | +92.5% |
| | HR@10 | 0.0208 | **0.0293** | +40.6% |
| | NDCG@10 | 0.0103 | **0.0159** | +53.9% |
| | Time/Epoch | 19.94s | +2.22s | |
| | Epochs | 100 | 150 | |
| Clothing | HR@1 | 0.0071 | **0.0171** | +139.1% |
| | HR@10 | 0.0360 | **0.0626** | +73.9% |
| | NDCG@10 | 0.0199 | **0.0377** | +89.1% |
| | Time/Epoch | 25.20s | +8.27s | |
| | Epochs | 400 | 300 | |

demands just a few more seconds. SEvo is arguably superior to these cumbersome GNN modules in real-world applications. Combining Table 2 and Table 3, it can be inferred that the performance gain increases as the number of items increases. This can be explained by the fact that the randomness of sampling leads to a much more inconsistent evolution when more and more nodes are encountered [56]. SEvo thus plays an increasingly important role as it is capable of imposing direct consistency constraints on embeddings.

Since SASRec is a pioneer in the field of sequential recommendation, it will serve as the default backbone for subsequent studies.

## 3.2 Empirical Analysis

**Convergence comparison.** In Section 2.3, we theoretically verified the necessity of rescaling the original Neumann series approximation for faster convergence. Figure 2a shows the loss curves of SASRec trained with AdamW under identical settings other than the form of SEvo. Without rescaling, SASRec exhibits significantly slower convergence, consistent with the conclusion in Theorem 2. While the theoretical worst-case convergence rate of the corrected SEvo is only 1% of the normal gradient descent when $\beta = 0.99$, its practical performance is much better. SASRec trained with SEvo-enhanced AdamW initially converges marginally slower and catches up in the final stage.

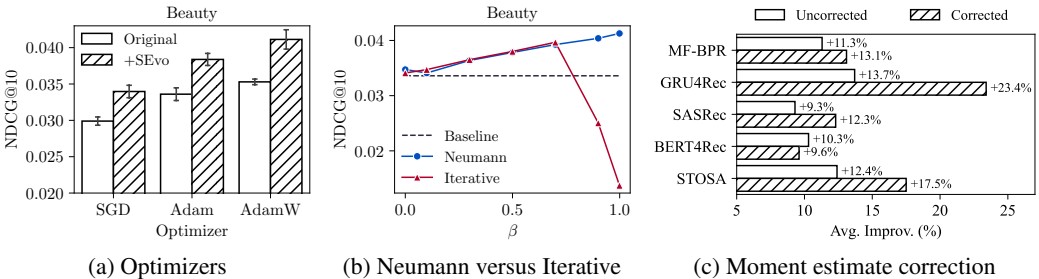

Figure 3: SEvo ablation experiments.

**Smoothness evaluation.** Figure 2b demonstrates the variation's smoothness throughout the evolution process and the eventual embedding differences from $\beta = 0$ to $\beta = 0.999$. **(I) → (II):** The original variations exhibit a similar degree of smoothness, but after transformation, they are quite different——smoother as $\beta$ increases. **(II) → (III):** Consequently, the embedding trained with a stronger smoothness constraint becomes smoother as well. The structure-aware embedding evolution successfully makes related nodes closer in the latent space. Although smoothness is not the sole quality measure of embedding, combined with the analyses above, we can conclude that SEvo injects appropriate structural information under the default setting of $\beta = 0.99$.

## 3.3 Ablation Study

**SEvo for various optimizers.** It is of interest to study whether SEvo can be extended to other commonly used optimizers such as SGD and Adam. Figure 3a compares NDCG@10 performance on Beauty and MovieLens-1M. For a fair comparison, the hyperparameters are tuned independently. It is evident that the performance of SGD, Adam, and AdamW improves significantly after integrating SEvo, with AdamW achieving the best as it does not need to smooth the weight decay.

**Neumann series approximation versus iterative approximation.** Theorem 1 suggests that the Neumann series approximation is preferable to the commonly used iterative approximation because the latter is not always direction-aware and thus a conservative hyperparameter of $\beta$ is needed to ensure convergence. This conclusion can also be drawn from Figure 3b. When only a little smoothness is required, their performance is comparable as both approximations differ only at the last term. The iterative approximation however fails to ensure convergence once $\beta > 0.7$ on the Beauty dataset, potentially resulting in a lack of smoothness.

**Moment estimate correction for AdamW.** We compare SEvo-enhanced AdamW with or without moment estimate correction in Figure 3c, in which average relative improvements against the baseline are presented for each recommender. Overall, the two variants of SEvo-enhanced AdamW perform comparably, significantly surpassing the baseline. However, in some cases (*e.g.*, GRU4Rec and STOSA), the moment estimate correction as suggested in Theorem 3 is particularly useful to improve performance. Recall that BERT4Rec is trained using the output softmax from a separate fully-connected layer that is fully updated at each step. This may explain why the correction has little effect on BERTRec. In conclusion, the results underscore the importance of the proposed modification in alleviating bias in moment estimates.

## 3.4 Applications of SEvo Beyond Interaction Data

We further explore the potential of applying SEvo to other types of prior knowledge. On the one hand, the category smoothness constraint can also be fulfilled through SEvo (see Appendix C.2), leading to progressively stronger clustering effects as $\beta$ increases. This provides compelling visual evidence of why SEvo is inherently structure-aware. On the other hand, SEvo is arguably an efficient tool for transferring embedding knowledge (see Appendix C.3). Notice that the learning of other modules cannot be guided in the same way, so SEvo alone is still inferior to state-of-the-art knowledge distillation methods [21, 55]. Fortunately, SEvo and other methods can work together to further boost the recommendation performance.

# 4 Related Work

**Recommender systems** are developed to enable users to quickly and accurately find relevant items in diverse applications, such as e-commerce [58], online news [15] and social media [5]. Typically, the entities involved are embedded into a latent space [6, 12, 54], and then decision models are built on top of the embedding for tasks like collaborative filtering [16] and context/knowledge-aware recommendation [46, 44]. Sequential recommendation [36, 25] focuses on capturing users' dynamic interests from their historical interaction sequences. Early approaches adapted recurrent neural networks (RNNs) [17] and convolutional filters [42] for sequence modeling. Recently, Transformer [45, 11] has become a popular architecture for sequence modeling due to its parallel efficiency and superior performance. SASRec [22] and BERT4Rec [40] use unidirectional and bidirectional self-attention, respectively. Fan et al. [13] proposed a novel stochastic self-attention (STOSA) to model the uncertainty of sequential behaviors.

**Graph neural networks** [2, 10] are a type of neural network designed to operate on graph-structured data, in concert with a weighted adjacency matrix to characterize the pairwise relations between nodes. GNN equipped with this adjacency matrix can be used for message passing between nodes. The most relevant work is the optimization framework proposed in [57] for solving semi-supervised learning problems via a smoothness constraint. This graph regularization approach has recently inspired a series of work [7, 28, 60]. As opposed to applying it to smooth node representations [24] or labels [19], it is employed here primarily to balance smoothness and convergence on the variation.

**Structural information** in recommendation is typically learned through GNN as well, with specific modifications made to cope with like data sparsity [52, 29]. LightGCN [16] is a pioneering collaborative filtering work on modeling user-item relations, which removes nonlinearities for easier training. To further utilize sequential information, previous efforts focus on equipping sequence models with complex GNN modules, but this inevitably increases the computational cost of training and inference, making it unappealing for practical recommendation. For example, SR-GNN [48] and LESSR [8] need dynamically construct adjacency matrices for each batch of sequences. Differently, MAERec [51] proposes an adaptive data augmentation to boost a novel graph masked autoencoder, which learns to sample less noisy paths from semantic similarity graph for subsequent reconstruction tasks. The resulting strong self-supervision signals help the model capture more useful information.

# 5 Broader Impact and Limitations

Utilizing both sequential and structural information is beneficial yet challenging, and SEvo proposed in this paper suggests a novel and effective technical route for this purpose. Compared to other explicit GNN modules, SEvo is light-weight and easy-to-use in practice. These insights may inspire future research efforts regarding structure-aware optimization. However, there are still some limitations. Firstly, the training-free nature of SEvo makes it versatile, but also limits the expressive power. For a specific task, a sophisticated GNN module may be more desirable for achieving higher recommendation accuracy. Secondly, it might not be so straightforward to apply SEvo to the scenario involving multiple types of prior knowledge. Some efforts [43, 33] in the field of multiple graph learning have proposed some technically feasible solutions. However, these approaches still encounter challenges in terms of efficiency, particularly in the context of recommendation systems.

# 6 Conclusion and Future Work

In this work, we have proposed a novel update mechanism for injecting graph structural information into embedding. Theoretical analyses of the convergence properties motivate some necessary modifications to the proposed method. SEvo can be seamlessly integrated into existing optimizers. For AdamW, we recorrect the moment estimates to make it more robust. Besides, an interesting direction for future work is extending SEvo to multiplex heterogeneous graphs [3], as real-world entities often participate in various relation networks. Furthermore, we believe that SEvo holds potential for application to dynamic graph structures and incremental updates [53]. Two challenges may be encountered in practice: the computational overhead associated with the ongoing adjacency matrix normalization, and how to adaptively weaken the outdated historical information.

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

# Contents

# A Proofs

## A.1 Proof of Theorem 1

In this part, we are to prove the structure-aware/direction-aware properties of the $L$-layer iterative approximation [57]:

$$\hat{\psi}_{iter}(\Delta\mathbf{E}) := \hat{\psi}_L(\Delta\mathbf{E}) = \underbrace{\left\{(1-\beta)\sum_{l=0}^{L-1}\beta^l\tilde{\mathbf{A}}^l + \beta^L\tilde{\mathbf{A}}^L\right\}}_{=:\mathbf{P}'}\Delta\mathbf{E},\tag{9}$$

and the $L$-layer Neumann series approximation [39]:

$$\hat{\psi}_{nsa}(\Delta\mathbf{E}) = \underbrace{(1-\beta)\sum_{l=0}^{L}\beta^l\tilde{\mathbf{A}}^l}_{=:\mathbf{P}}\Delta\mathbf{E}.\tag{10}$$

**Fact 1.** *If $L$ is odd, the geometric series $S(x) = \sum_{k=0}^{L}x^l$ is monotonically increasing when $x \leq 0$.*

*Proof.* It is easy to show that

$$S(x) = \frac{1-x^{L+1}}{1-x},$$

and the derivative w.r.t $x \neq 1$ is

$$S'(x) = \frac{Lx^{L+1} - (L+1)x^L + 1}{(1-x)^2}.$$

If $L$ is odd, $S'(x)$ is positive when $x \leq 0$ and in this case $S(x)$ is monotonically increasing. $\square$

**Lemma 1.** *Given a normalized adjacency matrix $\tilde{\mathbf{A}} \in \mathbb{R}^{n\times n}$, let the symmetric matrix deduced from the Neumann series approximation be*

$$\mathbf{P} = (1-\beta)\sum_{l=0}^{L}\beta^l\tilde{\mathbf{A}}^l.\tag{11}$$

*Denoted by $\lambda_{\min}(\mathbf{P}), \lambda_{\max}(\mathbf{P})$ the smallest and largest eigenvalues of $\mathbf{P}$, respectively, then we have $\forall \beta \in [0,1)$*

$$\frac{1-\beta}{1+\beta}(1-\beta^L) \leq \lambda_{\min}(\mathbf{P}) \leq \lambda_{\max}(\mathbf{P}) = 1 - \beta^{L+1}.\tag{12}$$

*Proof.* It is easy to shown that the eigenvalues of $\mathbf{P}$ are in the form of

$$\tilde{\lambda}_i = (1-\beta)\sum_{l=0}^{L}\beta^l\lambda_i^l, \ i = 1, 2, \ldots, n,\tag{13}$$

where $\lambda_1 \leq \lambda_2 \cdots \leq \lambda_n$ denote the eigenvalues of $\tilde{\mathbf{A}}$. Recall that these eigenvalues all fall into $[-1, 1]$ and $\lambda_n = 1$ can be achieved exactly [38]. Hence,

$$\tilde{\lambda}_n = (1-\beta)\sum_{l=0}^{L}\beta^l = 1 - \beta^{L+1}\tag{14}$$

is the largest eigenvalue of $\mathbf{P}$.

In addition, notice that the last term $(1 - \beta)\beta^L\lambda^L$ is non-negative when $L$ is even. Then, we can get a lower bound no matter $L$ is odd or even:

$$\tilde{\lambda} = (1 - \beta)\sum_{l=0}^{L}\beta^l\lambda^l \geq (1 - \beta)\underbrace{\sum_{l=0}^{2\lfloor(L-1)/2\rfloor+1}\beta^l\lambda^l}_{=:S(\lambda)}. \tag{15}$$

The minimum of $S(\lambda)$ must be achieved in $[-1, 0]$ because $S(-\lambda) \leq S(\lambda)$ for any $\lambda > 0$. In fact, in view of Fact 1, we know that $S(\lambda) \geq S(-1)$. Hence, we have

$$\begin{aligned}
\tilde{\lambda} &\geq (1 - \beta)S(-1) = (1 - \beta)\sum_{l=0}^{2\lfloor(L-1)/2\rfloor+1}\beta^l(-1)^l \\
&= (1 - \beta)\sum_{l=0}^{\lfloor(L-1)/2\rfloor}\left(\beta^{2l} - \beta^{2l+1}\right) \\
&= (1 - \beta)^2\sum_{l=0}^{\lfloor(L-1)/2\rfloor}\beta^{2l} = (1 - \beta)^2\frac{(1 - \beta^{2\lfloor(L-1)/2\rfloor+2})}{1 - \beta^2} \\
&= \frac{1 - \beta}{1 + \beta}(1 - \beta^{2\lfloor(L-1)/2\rfloor+2}) \geq \frac{1 - \beta}{1 + \beta}(1 - \beta^L).
\end{aligned} \tag{16}$$

The last inequality holds because $2\lfloor(L - 1)/2\rfloor + 2 \geq L$. Therefore, the smallest eigenvalue of $\mathbf{P}$ must be greater than

$$\frac{1 - \beta}{1 + \beta}(1 - \beta^L).$$

$\square$

**Proposition 2.** *The Neumann series approximation is structure-aware and direction-aware for any $\beta \in [0, 1)$.*

*Proof.* In view of Lemma 1, we know

$$\lambda_{\min}(\mathbf{P}) \geq \frac{1 - \beta}{1 + \beta}(1 - \beta^L) > 0, \quad \forall \beta \in [0, 1),$$

so $\mathbf{P}$ is positive definite and thus $\hat{\psi}_{nsa}(\cdot)$ is direction-aware. Also, notice that $\mathbf{P}$ has the same eigenvectors as $\tilde{\mathbf{A}}$, and so does $\tilde{\mathbf{L}}$. Hence, it is also structure-aware:

$$\begin{aligned}
\langle\hat{\psi}_{nsa}(\mathbf{x}), \tilde{\mathbf{L}}\hat{\psi}_{nsa}(\mathbf{x})\rangle &= \langle\hat{\psi}_{nsa}(\mathbf{x}), \tilde{\mathbf{L}}\hat{\psi}_{nsa}(\mathbf{x})\rangle = \langle\mathbf{P}\mathbf{x}, \tilde{\mathbf{L}}\mathbf{P}\mathbf{x}\rangle \\
&= \langle\mathbf{x}, \mathbf{P}^T\tilde{\mathbf{L}}\mathbf{P}\mathbf{x}\rangle \leq \langle\mathbf{x}, \tilde{\mathbf{L}}\mathbf{x}\rangle.
\end{aligned}$$

The last inequality follows from the fact $\lambda_{\max}(\mathbf{P}) \leq 1$.

$\square$

**Lemma 2.** *Given a normalized adjacency matrix $\tilde{\mathbf{A}} \in \mathbb{R}^{n\times n}$, let the symmetric matrix deduced from the iterative approximation be*

$$\mathbf{P}' = (1 - \beta)\sum_{l=0}^{L-1}\beta^l\tilde{\mathbf{A}}^l + \beta^L\tilde{\mathbf{A}}^L. \tag{17}$$

*We have $\lambda_{\min}(\mathbf{P}') > 0, \forall \beta < 1/2$.*

*Proof.* This conclusion is trivial for the case of $L \leq 1$. Let us assume that $L \geq 2$. Firstly, rewrite Eq. (17) as

$$\mathbf{P}' = \mathbf{P} + \beta^L\tilde{\mathbf{A}}^L, \tag{18}$$

where $\mathbf{P} := (1 - \beta) \sum_{l=0}^{L-1} \beta^l \tilde{\mathbf{A}}^l$. $\mathbf{P}$ is positive definite in view of Lemma 1 and $\beta^L \tilde{\mathbf{A}}^L$ is positive semidefinite when $L$ is even. Therefore, only the case of $L \geq 3$ needs to be proved. For a vector $\mathbf{x}$, we have

$$\mathbf{x}^T \mathbf{P}' \mathbf{x} = \mathbf{x}^T \mathbf{P} \mathbf{x} + \beta^L \mathbf{x}^T \tilde{\mathbf{A}}^L \mathbf{x} \geq \lambda_{\min}(\mathbf{P}) \|\mathbf{x}\|_2^2 + \beta^L \lambda_{\min}(\tilde{\mathbf{A}})^L \|\mathbf{x}\|_2^2$$

$$\geq \left( \frac{1 - \beta}{1 + \beta} (1 - \beta^{L-1}) + \beta^L \lambda_{\min}(\tilde{\mathbf{A}})^L \right) \|\mathbf{x}\|_2^2 \tag{19}$$

$$\geq \left( \frac{1 - \beta}{1 + \beta} (1 - \beta^{L-1}) - \beta^L \right) \|\mathbf{x}\|_2^2 = \frac{1 - \beta - \beta^{L-1} - \beta^{L+1}}{1 + \beta} \|\mathbf{x}\|_2^2$$

$$\geq \frac{1 - \beta - \beta^2 - \beta^4}{1 + \beta} \|\mathbf{x}\|_2^2. \tag{20}$$

The first two inequalities follow from Lemma 1. The last inequality holds by noting the fact that, for $L \geq 3$ and $\beta < 1/2$,

$$\beta + \beta^{L-1} + \beta^{L+1} \leq \beta + \beta^2 + \beta^4 < 13/16.$$

Therefore,

$$\lambda_{\min}(\mathbf{P}') = \min_{\mathbf{x}} \frac{\mathbf{x}^T \mathbf{P}' \mathbf{x}}{\|\mathbf{x}\|^2} \geq \frac{1 - \beta - \beta^2 - \beta^4}{1 + \beta} > 0. \tag{21}$$

$\square$

**Proposition 3.** *The iterative approximation is direction-aware for all possible normalized adjacency matrices and $L \geq 0$, if and only if $\beta < 1/2$.*

*Proof.* If $\beta < 1/2$, we have $\lambda_{\min}(\mathbf{P}') > 0$ according to Lemma 2, and thus $\hat{\psi}$ is direction-aware. Conversely, if $\beta \geq 1/2$, we can construct an adjacency matrix $\tilde{\mathbf{A}}$ such that $\lambda_{\min}(\mathbf{P}') \leq 0$ for some $L$. Let us assume that $L = 1$ and

$$\tilde{\mathbf{A}} := \begin{bmatrix} 0 & 1 \\ 1 & 0 \end{bmatrix}. \tag{22}$$

In this case, we have

$$\mathbf{P}' = (1 - \beta)\mathbf{I} + \beta\tilde{\mathbf{A}} = \begin{bmatrix} 1 - \beta & \beta \\ \beta & 1 - \beta \end{bmatrix}, \tag{23}$$

whose eigenvalues are 1 and $1 - 2\beta$. The latter is non-positive for any $\beta \geq 1/2$. $\square$

**Remark 2.** *The construction of $\tilde{\mathbf{A}}$ in Eq. (22) is not unique. In fact, any bipartite graph can be used as a counterexample.*

**Corollary 1** (The proof of Theorem 1). *The iterative approximation is direction-aware for all possible normalized adjacency matrices and $L \geq 0$, if and only if $\beta < 1/2$. In contrast, the Neumann series approximation*

$$\hat{\psi}_{nsa}(\Delta \mathbf{E}) = (1 - \beta) \sum_{l=0}^{L} \beta^l \tilde{\mathbf{A}}^l \Delta \mathbf{E}, \tag{24}$$

*is structure-aware and direction-aware for any $\beta \in [0, 1)$.*

*Proof.* This is a corollary of Proposition 2 and Proposition 3. $\square$

**Proposition 4.** *The rescaled Neumann series approximation $\hat{\psi}$ is structure-aware and direction-aware, and converges to the optimal solution as $L$ increases.*

*Proof.* The convergence to the optimal solution is obvious by noting that

$$\begin{aligned} \lim_{L \to +\infty} \hat{\psi}(\Delta \mathbf{E}) &= \lim_{L \to +\infty} \frac{1}{1 - \beta^{L+1}} \cdot \lim_{L \to +\infty} \hat{\psi}_{nsa}(\Delta \mathbf{E}) \\ &= 1 \cdot \psi^*(\Delta \mathbf{E}) = \psi^*(\Delta \mathbf{E}). \end{aligned} \tag{25}$$

Similar to Proposition 2, it can be proved that $\frac{1}{1 - \beta^{L+1}} \mathbf{P}$ is positive definite with a largest eigenvalue $\leq 1$. Therefore, the rescaled transformation is also structure-aware and direction-aware.

$\square$

## A.2 Proof of Theorem 2

Before delving into the proof of the convergence, we would like to claim that the lemmas below are well known and can be found in most textbooks [32, 1] on convex optimization. For the sake of completeness, we provide here these proofs. Hereinafter, we use $\|\mathbf{X}\|_2$ to denote the spectral norm which returns the largest singular value of the matrix $\mathbf{X}$.

**Lemma 3.** *For a twice continuously differentiable function $f : \mathbb{R}^n \rightarrow \mathbb{R}$ with $\|\nabla^2 f(\mathbf{x})\|_2 \leq C$, $\forall \mathbf{x} \in \mathbf{dom}(f)$, we have*

$$f(\mathbf{y}) \leq f(\mathbf{x}) + \langle \nabla f(\mathbf{x}), \mathbf{y} - \mathbf{x} \rangle + \frac{C}{2} \|\mathbf{y} - \mathbf{x}\|_2^2. \tag{26}$$

*Proof.* For a Taylor expansion of $f(\mathbf{x})$, there exists a $\mathbf{z} = \tau\mathbf{x} + (1 - \tau)\mathbf{y}$ for some $\tau \in [0, 1]$ such that

$$f(\mathbf{y}) = f(\mathbf{x}) + \langle \nabla f(\mathbf{x}), (\mathbf{y} - \mathbf{x}) \rangle + \frac{(\mathbf{y} - \mathbf{x})^T \nabla^2 f(\mathbf{z})(\mathbf{y} - \mathbf{x})}{2}$$
$$\leq f(\mathbf{x}) + \langle \nabla f(\mathbf{x})^T (\mathbf{y} - \mathbf{x}) \rangle + \frac{C}{2} \|\mathbf{y} - \mathbf{x}\|_2^2.$$

$\square$

**Lemma 4.** *For a positive definite matrix $\mathbf{P}$, let $\|\mathbf{x}\|_{\mathbf{P}} := (\mathbf{x}^T \mathbf{P} \mathbf{x})^{1/2}$ be the quadratic norm induced from $\mathbf{P}$. If the eigenvalues of $\mathbf{P}$ fall into $[a, b]$, then*

$$\|\mathbf{P}\mathbf{x}\|_2^2 \leq b\|\mathbf{x}\|_{\mathbf{P}}^2, \quad a\|\mathbf{x}\|_2^2 \leq \|\mathbf{x}\|_{\mathbf{P}}^2. \tag{27}$$

*Proof.* Firstly,

$$\|\mathbf{P}\mathbf{x}\|_2 = \|\mathbf{P}^{1/2}\mathbf{P}^{1/2}\mathbf{x}\|_2 \leq \|\mathbf{P}^{1/2}\|_2 \|\mathbf{x}\|_{\mathbf{P}} \leq \sqrt{b}\|\mathbf{x}\|_{\mathbf{P}}.$$

Secondly,

$$\|\mathbf{x}\|_2 = \|\mathbf{P}^{-1/2}\mathbf{P}^{1/2}\mathbf{x}\|_2 \leq \|\mathbf{P}^{-1/2}\|_2 \|\mathbf{x}\|_{\mathbf{P}} \leq \frac{1}{\sqrt{a}}\|\mathbf{x}\|_{\mathbf{P}}.$$

$\square$

**Theorem 4.** *Let $f : \mathbb{R}^n \times \mathbb{R}^m \rightarrow \mathbb{R}$ be a twice continuously differentiable function bounded below, and its Hessian matrix satisfies*

$$\|\nabla^2 f(\mathbf{x}, \mathbf{y})\|_2 \leq C, \quad \forall(\mathbf{x}, \mathbf{y}) \in \mathbf{dom}(f) \tag{28}$$

*for some constant $C$. The following gradient descent scheme is used to train $\mathbf{x}, \mathbf{y}$:*

$$\mathbf{x}_{t+1} \leftarrow \mathbf{x}_t - \eta\mathbf{P}\nabla_{\mathbf{x}}f(\mathbf{x}_t, \mathbf{y}_t), \quad \mathbf{y}_{t+1} \leftarrow \mathbf{y}_t - \eta'\nabla_{\mathbf{y}}f(\mathbf{x}_t, \mathbf{y}_t), \tag{29}$$

*where $\mathbf{P}$ is deduced from the $L$-layer Neumann series approximation. Then, after $T$ updates, we have,*

$$\min_{t \leq T} \|\nabla f(\mathbf{x}, \mathbf{y})\|_2^2 \leq \frac{2C}{\gamma(T+1)}\epsilon, \tag{30}$$

*where $\epsilon = f(\mathbf{x}, \mathbf{y}) - f(\mathbf{x}^*, \mathbf{y}^*)$ and*

$$\gamma = \begin{cases} \frac{1-\beta}{1+\beta}(1 - \beta^L)(1 + \beta^{L+1}), & \text{if } \eta = \eta' = \frac{1}{C} \\ \frac{1-\beta}{1+\beta}\frac{1-\beta^L}{1-\beta^{L+1}}, & \text{otherwise} \end{cases}.$$

*Proof.* The update formula (29) can be unified into

$$\mathbf{z}_{t+1} := \begin{bmatrix} \mathbf{x}_{t+1} \\ \mathbf{y}_{t+1} \end{bmatrix} = \begin{bmatrix} \mathbf{x}_t \\ \mathbf{y}_t \end{bmatrix} - \underbrace{\begin{bmatrix} \eta\mathbf{P} & 0 \\ 0 & \eta'\mathbf{I} \end{bmatrix}}_{=:\tilde{\mathbf{P}}} \nabla f(\mathbf{x}_t, \mathbf{y}_t).$$

It is easy to show that

$$a := \min(\tfrac{1-\beta}{1+\beta}(1 - \beta^L)\eta, \eta') \leq \lambda_{\min}(\tilde{\mathbf{P}})$$
$$\leq \lambda_{\max}(\tilde{\mathbf{P}}) = \max((1 - \beta^{L+1})\eta, \eta') =: b.$$

Specifically,

$$a = \begin{cases} \frac{1-\beta}{1+\beta}(1 - \beta^L)\frac{1}{C}, & \text{if } \eta = \eta' = \frac{1}{C} \\ \frac{1-\beta}{1+\beta}\frac{1-\beta^L}{1-\beta^{L+1}}\frac{1}{C}, & \text{otherwise} \end{cases},$$

$$b = \begin{cases} (1 - \beta^{L+1})\frac{1}{C}, & \text{if } \eta = \eta' = \frac{1}{C} \\ \frac{1}{C}, & \text{otherwise} \end{cases}.$$

In view of Lemma 3 and Lemma 4, we have

$$f(\mathbf{z}_{t+1}) \leq f(\mathbf{z}_t) - \langle \nabla f(\mathbf{z}_t), \tilde{\mathbf{P}}\nabla f(\mathbf{z}_t) \rangle + \frac{C}{2}\|\tilde{\mathbf{P}}\nabla f(\mathbf{z}_t)\|_2^2$$

$$= f(\mathbf{z}_t) - \|\nabla f(\mathbf{z}_t)\|_{\tilde{\mathbf{P}}}^2 + \frac{C}{2}\|\tilde{\mathbf{P}}\nabla f(\mathbf{z}_t)\|_2^2$$

$$\leq f(\mathbf{z}_t) - \|\nabla f(\mathbf{z}_t)\|_{\tilde{\mathbf{P}}}^2 + \frac{bC}{2}\|\nabla f(\mathbf{z}_t)\|_{\tilde{\mathbf{P}}}^2 \tag{31}$$

$$= f(\mathbf{z}_t) - (1 - \frac{bC}{2})\|\nabla f(\mathbf{z}_t)\|_{\tilde{\mathbf{P}}}^2$$

$$\leq f(\mathbf{z}_t) - a(1 - \frac{bC}{2})\|\nabla f(\mathbf{z}_t)\|_2^2. \tag{32}$$

Denoted by

$$\gamma \quad := aC(2 - bC)$$
$$= \begin{cases} \frac{1-\beta}{1+\beta}(1 - \beta^L)(1 + \beta^{L+1}), & \text{if } \eta = \eta' = \frac{1}{C} \\ \frac{1-\beta}{1+\beta}\frac{1-\beta^L}{1-\beta^{L+1}}, & \text{otherwise} \end{cases}, \tag{33}$$

we have

$$\min_{0 \leq t \leq T} \|\nabla f(\mathbf{x}_t, \mathbf{y}_t)\|_2^2 \leq \frac{1}{T+1}\sum_{t=0}^{T}\|\nabla f(\mathbf{x}_t, \mathbf{y}_t)\|_2^2$$

$$\leq \frac{1}{T+1}\sum_{t=0}^{T}\frac{2C}{\gamma}(f(\mathbf{x}_t, \mathbf{y}_t) - f(\mathbf{x}_{t+1}, \mathbf{y}_{t+1}))$$

$$= \frac{2C}{\gamma(T+1)}(f(\mathbf{x}_0, \mathbf{y}_0) - f(\mathbf{x}_{T+1}, \mathbf{y}_{T+1}))$$

$$\leq \frac{2C}{\gamma(T+1)}\epsilon.$$

$\square$

**Theorem 5** (The proof of Theorem 2). *If $\eta = \eta' = 1/C$, after $T$ updates, we have*

$$\min_{t \leq T}\|\nabla \mathcal{L}(\mathbf{E}_t, \boldsymbol{\theta}_t)\|_2^2 = \mathcal{O}\big(C/((1 - \beta)^2 T)\big). \tag{34}$$

*If we adopt a modified learning rate for embedding:*

$$\eta = \frac{1}{(1 - \beta^{L+1})C}, \tag{35}$$

*the convergence rate could be improved to $\mathcal{O}\big(C/((1 - \beta)T)\big)$.*

*Proof.* This is true for $L = 0$, since in this case the update mechanism becomes a normal gradient descent regardless of $\eta = 1/C$ or $\eta = \frac{1}{(1-\beta)C}$. Let us prove a general case for $L \geq 1$ next.

According to Theorem 4, we have

$$\min_{t \leq T} \|\nabla \mathcal{L}(\mathbf{E}_t, \boldsymbol{\theta}_t)\|_2^2 \leq \frac{2C}{\gamma(T+1)} \epsilon,$$

where $\epsilon = \mathcal{L}(\mathbf{E}, \boldsymbol{\theta}) - \mathcal{L}(\mathbf{E}^*, \boldsymbol{\theta}^*)$ and

$$\gamma = \begin{cases} \frac{1-\beta}{1+\beta}(1-\beta^L)(1+\beta^{L+1}), & \text{if } \eta = \eta' = \frac{1}{C} \\ \frac{1-\beta}{1+\beta}\frac{1-\beta^L}{1-\beta^{L+1}}, & \text{otherwise} \end{cases}.$$

Notice that, for $\eta = \eta' = 1/C$,

$$\begin{aligned} \lim_{\beta \to 1} \frac{1/\gamma}{1/(1-\beta)^2} &= \lim_{\beta \to 1} \frac{\frac{1}{\frac{1-\beta}{1+\beta}(1-\beta^L)(1+\beta^{L+1})}}{\frac{1}{(1-\beta)^2}} \\ &= \lim_{\beta \to 1} \frac{1-\beta}{1-\beta^L} = \frac{1}{L}, \end{aligned} \tag{36}$$

and for $\eta = \frac{1}{(1-\beta^{L+1})C}, \eta' = \frac{1}{C}$,

$$\begin{aligned} \lim_{\beta \to 1} \frac{1/\gamma}{1/(1-\beta)} &= \lim_{\beta \to 1} \frac{\frac{1}{\frac{1-\beta}{1+\beta}\frac{(1-\beta^L)}{(1-\beta^{L+1})}}}{\frac{1}{1-\beta}} \\ &= 2 \cdot \lim_{\beta \to 1} \frac{1-\beta^{L+1}}{1-\beta^L} = \frac{2(L+1)}{L}. \end{aligned} \tag{37}$$

Therefore,

$$\frac{1}{\gamma} = \begin{cases} \mathcal{O}(1/(1-\beta)^2), & \text{if } \eta = \eta' = \frac{1}{C} \\ \mathcal{O}(1/(1-\beta)), & \text{if } \eta = \frac{1}{(1-\beta^{K+1})C}, \eta' = \frac{1}{C} \end{cases}.$$

The remainder of the proof is straightforward.

$\square$

## A.3 Proofs of Proposition 1 and Theorem 3

Adam(W) [23] uses the bias-corrected moment estimates for updating because they are unbiased when the actual moments are stationary throughout the training. Below, Lemma 5 formally elaborates on this, and Theorem 6 extends Theorem 3 with a proof of unbiasedness.

**Lemma 5** ([23]). *Denoted by $\hat{\mathbf{m}}_t = \mathbf{m}_t/(1-\beta_1^t)$ and $\hat{\mathbf{v}}_t = \mathbf{v}_t/(1-\beta_2^t)$ the bias-corrected estimates, if the first and second moments are stationary, i.e.,*

$$\mathbb{E}[\mathbf{g}_t] = \mathbb{E}[\mathbf{g}], \quad \mathbb{E}[\mathbf{g}_t^2] = \mathbb{E}[\mathbf{g}^2], \quad \forall t = 1, 2, \dots,$$

*then these bias-corrected estimates are unbiased:*

$$\mathbb{E}[\hat{\mathbf{m}}_t] = \mathbb{E}[\mathbf{g}], \quad \mathbb{E}[\hat{\mathbf{v}}_t] = \mathbb{E}[\mathbf{g}^2], \quad \forall t = 1, 2, \dots.$$

**Proposition 5.** *If a node is no longer sampled in subsequent $p$ batches after step $t$, we have*

$$\Delta \mathbf{e}_{t+p-1} = \kappa \cdot \frac{\beta_1^p}{\sqrt{\beta_2^p}} \Delta \mathbf{e}_{t-1}, \tag{38}$$

*where the coefficient of $\kappa$ is mainly determined by $t$.*

*Proof.* In this case, the iterative formula becomes

$$\mathbf{m}_{t+j} = \beta_1^j \mathbf{m}_t + \mathbf{0}, \quad \mathbf{v}_{t+j} = \beta_2^j \mathbf{v}_t + \mathbf{0}, \quad \forall j = 0, 1, \dots, p.$$

Therefore,

$$\Delta\mathbf{e}_{t+p-1} = \frac{\hat{\mathbf{m}}_{t+p}}{\sqrt{\hat{\mathbf{v}}_{t+p}}} = \frac{\sqrt{1 - \beta_2^{t+p}}}{1 - \beta_1^{t+p}} \frac{\mathbf{m}_{t+p}}{\sqrt{\mathbf{v}_{t+p}}}$$

$$= \frac{\sqrt{1 - \beta_2^{t+p}}}{1 - \beta_1^{t+p}} \frac{\beta_1^p}{\sqrt{\beta_2^p}} \frac{\mathbf{m}_t}{\sqrt{\mathbf{v}_t}} = \frac{\beta_1^p \sqrt{1 - \beta_2^{t+p}}}{\sqrt{\beta_2^p}(1 - \beta_1^{t+p})} \frac{\mathbf{m}_t}{\sqrt{\mathbf{v}_t}}$$

$$= \frac{\beta_1^p (1 - \beta_1^t)\sqrt{1 - \beta_2^{t+p}}}{\sqrt{\beta_2^p}(1 - \beta_1^{t+p})\sqrt{1 - \beta_2^t}} \frac{\hat{\mathbf{m}}_t}{\sqrt{\hat{\mathbf{v}}_t}}$$

$$= \frac{\beta_1^p}{\sqrt{\beta_2^p}} \cdot \underbrace{\frac{(1 - \beta_1^t)\sqrt{1 - \beta_2^{t+p}}}{(1 - \beta_1^{t+p})\sqrt{1 - \beta_2^t}}}_{=:\kappa} \Delta\mathbf{e}_{t-1}.$$

It is easy to show that

$$\lim_{t\to+\infty} \kappa(t,p) = 1, \quad \forall \beta_1, \beta_2 \in [0,1). \tag{39}$$

$\square$

**Theorem 6** (The proof of Theorem 3). *Under the same assumptions as in Lemma 5 and Proposition 1, the bias-corrected estimates are unbiased and $\Delta\mathbf{e}_{t+p-1} = \Delta\mathbf{e}_{t-1}$ if the estimates are updated in the following manner when $\mathbf{g}_t = \mathbf{0}$,*

$$\mathbf{m}_t = \beta_1 \mathbf{m}_{t-1} + (1 - \beta_1)\frac{1}{1 - \beta_1^{t-1}}\mathbf{m}_{t-1}, \quad \mathbf{v}_t = \beta_2 \mathbf{v}_{t-1} + (1 - \beta_2)\frac{1}{1 - \beta_2^{t-1}}\mathbf{v}_{t-1}. \tag{40}$$

*Proof.* We first show the unbiasedness of $\mathbf{m}_t$ and the proof for $\mathbf{v}_t$ is completely analogous. It remains only to show that

$$\mathbb{E}[\mathbf{m}_t] = (1 - \beta_1^t)\mathbb{E}[\mathbf{g}].$$

This is trivial for $t = 0$. Assuming that this also holds in the case of $t - 1$, it can be proved by induction.

If $\mathbf{g}_t \neq \mathbf{0}$,

$$\mathbb{E}[\mathbf{m}_t] = \beta_1 \mathbb{E}[\mathbf{m}_{t-1}] + (1 - \beta_1)\mathbb{E}[\mathbf{g}_t]$$
$$= \beta_1(1 - \beta_1^{t-1})\mathbb{E}[\mathbf{g}] + (1 - \beta_1)\mathbb{E}[\mathbf{g}]$$
$$= (1 - \beta_1^t)\mathbb{E}[\mathbf{g}].$$

If $\mathbf{g}_t = \mathbf{0}$,

$$\mathbb{E}[\mathbf{m}_t] = \beta_1 \mathbb{E}[\mathbf{m}_{t-1}] + (1 - \beta_1)\frac{1}{1 - \beta_1^{t-1}}\mathbb{E}[\mathbf{m}_{t-1}]$$
$$= \frac{1 - \beta_1^t}{1 - \beta_1^{t-1}}\mathbb{E}[\mathbf{m}_{t-1}] = (1 - \beta_1^t)\mathbb{E}[\mathbf{g}].$$

If the node is no longer sampled in subsequent $p$ batches after step $t$, we have

$$\mathbf{m}_{t+j} = \beta_1 \mathbf{m}_{t+j-1} + (1 - \beta_1)\frac{1}{1 - \beta_1^{t+j-1}}\mathbf{m}_{t+j-1}$$
$$= \frac{1 - \beta_1^{t+j}}{1 - \beta_1^{t+j-1}}\mathbf{m}_{t+j-1} = \frac{1 - \beta_1^{t+j}}{1 - \beta_1^{t+j-1}}\frac{1 - \beta_1^{t+j-1}}{1 - \beta_1^{t+j-2}}\mathbf{m}_{t+j-2}$$
$$= \cdots = \frac{1 - \beta_1^{t+j}}{1 - \beta_1^t}\mathbf{m}_t, \quad \forall j = 1, 2, \ldots, p.$$

Analogously, we have,

$$\mathbf{v}_{t+j} = \frac{1-\beta_2^{t+j}}{1-\beta_2^t}\mathbf{v}_t, \quad \forall j = 1, 2, \dots, p.$$

Therefore, the conclusion can be deduced from

$$\hat{\mathbf{m}}_{t+p} = \frac{1}{1-\beta_1^{t+j}}\mathbf{m}_{t+p} = \frac{1}{1-\beta_1^t}\mathbf{m}_t = \hat{\mathbf{m}}_t,$$

$$\hat{\mathbf{v}}_{t+p} = \frac{1}{1-\beta_2^{t+j}}\mathbf{v}_{t+p} = \frac{1}{1-\beta_2^t}\mathbf{v}_t = \hat{\mathbf{v}}_t.$$

$\square$

## A.4 Connection between LightGCN and SEvo-enhanced MF-BPR

For a $L$-layer LightGCN, it can be formulated as follows

$$\mathbf{F} = \psi(\mathbf{E}) := \sum_{l=0}^{L} \alpha_l \tilde{\mathbf{A}}^l \mathbf{E},$$

where $\alpha_l, l = 0, \dots, L$ represent the layer weights. According to the linear nature of the gradient operator, it can be obtained that

$$\nabla_{\mathbf{E}}\mathcal{L} = \psi(\nabla_{\mathbf{F}}\mathcal{L}).$$

Hence, denoted by $\zeta(\cdot)$ the gradient processing procedure of an optimizer, we can establish that LightGCN is identical to the following system:

$$
\begin{aligned}
\mathbf{F}(t) &= \psi(\mathbf{E}(t)) \\
&= \psi(\mathbf{E}(t-1) - \eta\Delta\mathbf{E}(t-1)) \\
&= \psi(\mathbf{E}(t-1)) - \eta\psi(\Delta\mathbf{E}(t-1)) \\
&= \mathbf{F}(t-1) - \eta\psi \circ \zeta \circ \psi(\nabla_{\mathbf{F}}\mathcal{L}).
\end{aligned}
$$

When $\zeta(\cdot)$ is an identity mapping (i.e., standard gradient descent), LightGCN is equivalent to MF-BPR with SEvo being applied twice at each update. However, when $\zeta(\cdot)$ is not an identity mapping (e.g., an optimizer with momentum or weight decay is integrated), they cannot be unified into a single system. Compared to explicit GNNs, SEvo is easy-to-use and has minimal impact on the forward pass, making it more suitable for assisting recommenders in simultaneously utilizing multiple types of information. These connections in part justify why SEvo can inject structural information directly.

## B Detailed Settings

### B.1 Algorithms

We present the algorithms of SEvo-enhanced Adam and SGD in 2 and Algorithm 3, respectively.

### B.2 Datasets

In this study, we perform experiments on six public datasets. Specifically, the Beauty, Toys, and Tools datasets are extracted from Amazon reviews published in 2014[2], while Electronics and Clothing are sourced from Amazon reviews published in 2018[3]. Additionally, the MovieLens-1M dataset is made available by GroupLens[4].

---

[2] https://cseweb.ucsd.edu/~jmcauley/datasets/amazon/links.html
[3] https://cseweb.ucsd.edu/~jmcauley/datasets/amazon_v2
[4] https://grouplens.org/datasets/movielens/1m

**Algorithm 2:** Adam enhanced by SEvo. Differences from the original Adam are colored in blue. The matrix operation below are element-wise.

**Input:** embedding matrix $\mathbf{E}$, learning rate $\eta$, momentum factors $\beta_1, \beta_2, \beta \in [0, 1)$, weight decay $\lambda$.

**foreach** *step t* **do**

$\mathbf{G}_t \leftarrow \nabla_{\mathbf{E}}\mathcal{L} + \lambda\mathbf{E}_{t-1}$ ;                                       // Get gradients

Update first/second moment estimates:

$$\mathbf{M}_t \leftarrow \beta_1\mathbf{M}_{t-1} + (1 - \beta_1)\mathbf{G}_t,$$
$$\mathbf{V}_t \leftarrow \beta_1\mathbf{V}_{t-1} + (1 - \beta_2)\mathbf{G}_t^2;$$

Compute bias-corrected first/second moment estimates:

$$\hat{\mathbf{M}}_t \leftarrow \mathbf{M}_t/(1 - \beta_1^t),$$
$$\hat{\mathbf{V}}_t \leftarrow \mathbf{V}_t/(1 - \beta_2^t);$$

Update via SEvo:

$$\mathbf{E}_t \leftarrow \mathbf{E}_{t-1} - \eta\,\hat{\psi}\left(\hat{\mathbf{M}}_t/\sqrt{\hat{\mathbf{V}}_t + \epsilon}; \beta\right).$$

**Output:** optimized embeddings $\mathbf{E}$.

---

**Algorithm 3:** SGD with momentum enhanced by SEvo. Differences from the original SGD are colored in blue. The matrix operation below are element-wise.

**Input:** embedding matrix $\mathbf{E}$, learning rate $\eta$, momentum factors $\mu, \beta \in [0, 1)$, weight decay $\lambda$.

**foreach** *step t* **do**

$\mathbf{G}_t \leftarrow \nabla_{\mathbf{E}}\mathcal{L} + \lambda\mathbf{E}_{t-1}$ ;                                       // Get gradients

$\mathbf{M}_t \leftarrow \mu\mathbf{M}_{t-1} + \mathbf{G}_t$ ;                                          // Moment update

Update via SEvo:

$$\mathbf{E}_t \leftarrow \mathbf{E}_{t-1} - \eta\,\hat{\psi}\left(\mathbf{M}_t; \beta\right).$$

**Output:** optimized embeddings $\mathbf{E}$.

---

### B.3 Baselines

Four GNN-based baselines for performance and efficiency benchmarks:

- **LightGCN** [16] is a pioneering collaborative filtering model that simplifies graph convolutional networks (GCNs) by removing nonlinearities for easier training. It uses only graph structural information and has no access to sequential information.

- **SR-GNN** [48] and **LESSR** [8] are two baselines dynamically constructing session graph. The former employs a gated graph neural network to obtain the final node vectors, while the latter utilizes edge-order preserving multigraph and a shortcut graph to address the lossy session encoding and ineffective long-range dependency capturing problems, respectively.

- **MAERec** [51] learns to sample less noisy paths from a semantic similarity graph for subsequent reconstruction tasks. However, we found that the official implementation treats the recommendation loss and reconstruction loss equally, leading to poor performance here. Therefore, an additional weight is attached to the reconstruction loss and a grid search is performed in the range of [0, 1]. Almost all hyperparameters are tuned for competitive performance.

Four sequence backbones to validate the effectiveness of SEvo:

- **GRU4Rec** [17] applies RNN [9] to recommendation with specific modifications made to cope with data sparsity. In addition to the learning rate in {1e-4, 5e-4, 1e-3, 5e-3} and weight decay in [0, 0.1], we also tune the dropout rate for node features in the range of [0, 0.7].

- **SASRec** [22] and **BERT4Rec** [40] are two pioneering works on sequential recommendation equipped with unidirectional and bidirectional self-attention, respectively. For BERT4Rec which employs a separate fully-connected layer for scoring, the weight matrix therein will also be smoothed by SEvo. In addition to some basic hyperparameters, the mask ratio is also researched for BERT4Rec.

- **STOSA** [13] is one of the state-of-the-art models. It aims to capture the uncertainty of sequential behaviors by modeling each item as a Gaussian distribution. The hyperparameters involved are tuned similarly to SASRec.

Four knowledge distillation methods used in Appendix C.3:

- **KD** [18] and **DKD** [55] are two logits-based approaches to transfer knowledge. DKD decomposes the classical KD loss into target class knowledge distillation loss and non-target class knowledge distillation loss.

- **RKD** [34] and **HTD** [21] are two ranking-based approaches. The former focuses the distillation of relational knowledge through distance-wise and angle-wise alignments, while the latter emphasizes the distillation of hierarchical topology by dividing nodes into multiple groups and requiring intra-group and inter-group alignments.

## C    Applications of SEvo beyond Interaction Data

Here, we preliminarily explore the exploitation of more types of knowledge besides consecutive occurrences. We first investigate some elementary factors for interaction data, and then introduce the applications of SEvo to node categories and knowledge distillation.

### C.1    Pairwise Similarity Estimation Factors

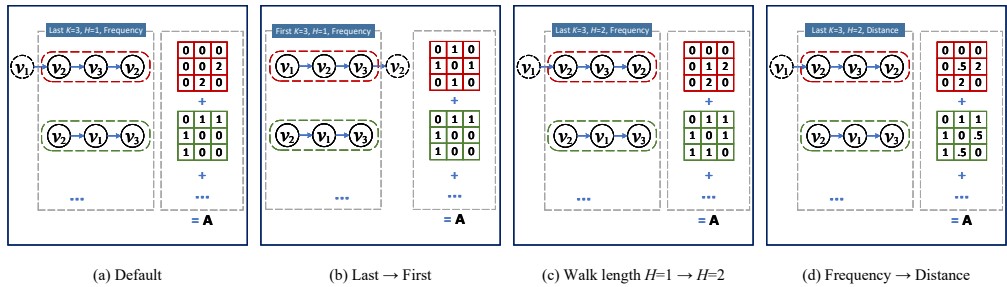

(a) Default          (b) Last → First          (c) Walk length $H$=1 → $H$=2          (d) Frequency → Distance

Figure 4: Illustrations of different pairwise similarity estimation methods based on interaction data. (a) The default is to adopt the co-occurrence frequency within the last $K$ items. (b) Using only the first $K$ items. (c) Allowing a maximum walk length of $H$ beyond 1. (d) Frequency-based similarity versus distance-based similarity.

Recent GNN-based sequence models [48, 49], as well as the SEvo-enhanced models reported in Section 3, estimate the pairwise similarity $w_{ij}$ between items $v_i$ and $v_j$ based on their co-occurrence frequency in sequences. In other words, items that appear consecutively more frequently are assumed more related. Yet there are some factors that deserve a closer look: **(1)** The maximum sequence length $K$ for construction to investigate the number of interactions required for accurate estimation; **(2)** Using only the first $K$ versus last $K$ interactions in each sequence to compare the utility of early and recent preferences; **(3)** Allowing related items to be connected by a walk of length $\leq H$ rather than strict consecutive occurrences; **(4)** Frequency-based similarity versus distance-based similarity. The former weights all related pairs equally, while the latter weights inversely to their walk length. For example, given a sequence $v_2 \rightarrow v_1 \rightarrow v_3$ with a maximum walk length of $H = 2$,

**Algorithm 4:** Python-style algorithm for similarity estimation based on interaction data.

```python
for seq in seqs:
    if first:
      seq = seq[:K] # First K items
    else:
      seq = seq[-K:] # Last K items
    for i in range(len(seq) - 1):
        # Maximum walk length H
        for h, j in enumerate(
            range(i + 1, min(i + H + 1, len(seq))),
            start=1
        ):
            if frequency: # Frequency
              A[seq[i], seq[j]] += 1
              A[seq[j], seq[i]] += 1
            else: # Distance
              A[seq[i], seq[j]] += 1 / h
              A[seq[j], seq[i]] += 1 / h
```

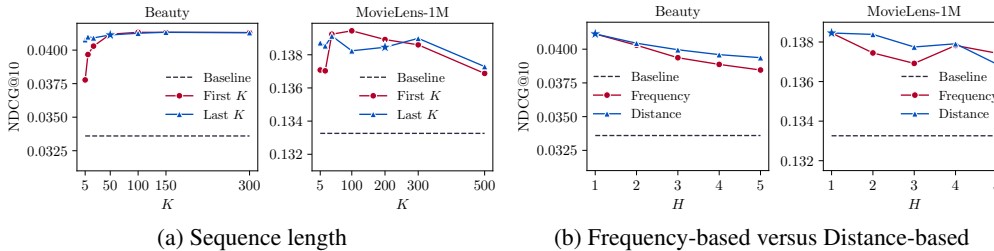

(a) Sequence length  (b) Frequency-based versus Distance-based

Figure 5: Comparison of similarity estimation across four potential factors. '⋆' indicates the default way applied to SEvo-enhanced sequence models in Section 3.1. (a) Using only the first/last $K$ items for pairwise similarity estimation. (b) Frequency- and distance-based similarity with a maximum walk length of $H$.

the frequency-based similarity of $(v_2, v_3)$ gives 1, while the distance-based similarity is $1/2$ (as the walk length from $v_2$ to $v_3$ is 2).

Figure 4 illustrates these four variants and Algorithm 4 details a step-by-step process. We further compare these four potential factors in Figure 5:

- Figure 5a shows the effect of confining the maximum sequence length to the first/last $K$ items, so only the early/recent preferences will be considered. In contrast to early interactions, recent ones imply more precise pairwise relations for future prediction, even for small $K$. With the increase of the maximum sequence length, the recommendation performance on Beauty improves steadily, but not the case for MovieLens-1M. This suggests that shopping relations may be more consistent than movie preferences.

- Figure 5b explores the relations beyond strict consecutive occurrences; that is, two items are considered related once they co-occur within a path of length $\leq H$. For the shopping and movie datasets, estimating similarity beyond co-occurrence frequency appears less reliable overall. We also compare frequency-based similarity with distance-based similarity that decreases weights for more distant pairs. It is clear that the distance-based approach performs more stably as the maximum walk length $H$ increases.

## C.2 SEvo for Intra-class Representation Proximity

Sometimes embeddings are expected to be smooth w.r.t. a prior knowledge. For example, in addition to the interaction data, each movie in the MovieLens-1M dataset is associated with at least one genre.

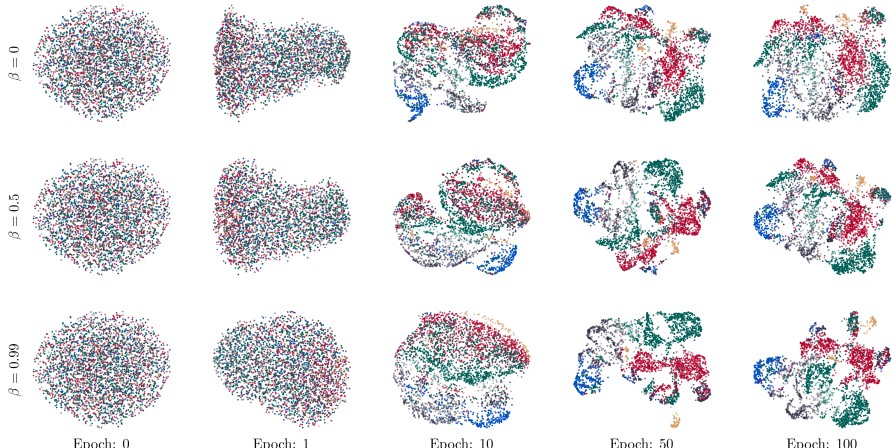

Figure 6: UMAP [30] visualization of movies based on their embeddings. For ease of differentiation, we group the 18 genres into 6 categories and colored them individually: Thriller/Crime/Action/Adventure; Horror/Mystery/Film-Noir; War/Drama/Romance; Comedy/Musical/Children's/Animation; Fantasy/Sci-Fi; Western/Documentary.

Table 4: Pairwise similarity estimation based on interaction data versus node categories (movie genres).

|  | | | MovieLens-1M | |
| --- | --- | --- | --- | --- |
|  | $\beta$ | HR@1 | HR@10 | NDCG@10 |
| Baseline (SASRec) | 0 | 0.0457 | 0.2482 | 0.1315 |
| Interaction data | 0.5 | 0.0494 | 0.2538 | 0.1362 |
|  | 0.99 | **0.0517** | **0.2567** | **0.1385** |
| Movie genres | 0.5 | 0.0492 | 0.2527 | 0.1352 |
|  | 0.99 | 0.0508 | 0.2549 | 0.1371 |

It is natural to assume that movies of the same genre are related to each other. Heuristically, we can define the similarity $w_{ij}$ to be 1 if $v_i$ and $v_j$ belong to the same genre and 0 otherwise.

As can be seen in Figure 6, such smoothness constraint can also be fulfilled through SEvo, leading to progressively stronger clustering effects as $\beta$ increases. However, the resulting performance gains are slightly less than those based on interaction data (see Table 4). One possible reason is that the movie genres are too coarse to provide particularly useful information. In conclusion, while smoothness is an appealing inductive bias, its utility depends on how well the imposed structural information agrees with the performance metrics of interest.

### C.3 SEvo for Knowledge Distillation

In addition to the affinity matrix extracted from interaction data or relation data, the pairwise similarity can also be estimated from a heavy-weight teacher model. Recall that Knowledge Distillation (KD) [18] encourages a light-weight student model to mimic the behaviors (*e.g.*, output distribution) of the teacher model, so the learned student model achieves both accuracy and efficiency. In general, higher-dimensional embeddings are capable of better fitting the underlying distribution between entities. The pairwise similarities extracted from a teacher model, needless to say, can be used to guide the embedding evolution of a student model. Unlike interaction or relation data, the deduced graph is dense if only the distance function is applied. Therefore, some graph construction steps including sparsification and reweighting should be involved as well. We attempt to use the widely used KNN graph here, and leave a more comprehensive study of graph construction [20] as a future work.

Table 5: Knowedge distillation from Teacher (SASRec with a embedding size of 200) to Student (SASRec with a embedding size of 20). The results are averaged over 5 independent runs on the Beauty dataset. 10-nearest neighbors (*i.e.*, $K = 10$) are selected for each node.

| | HR@1 | HR@5 | HR@10 | NDCG@5 | NDCG@10 |
|---|---|---|---|---|---|
| Teacher | 0.0198 | 0.0544 | 0.0786 | 0.0374 | 0.0452 |
| Student | 0.0094 | 0.0327 | 0.0526 | 0.0210 | 0.0275 |
| +KD [18] | 0.0105 | 0.0352 | 0.0552 | 0.0229 | 0.0294 |
| +RKD [34] | 0.0082 | 0.0311 | 0.0515 | 0.0196 | 0.0262 |
| +HTD [21] | 0.0085 | 0.0344 | 0.0549 | 0.0215 | 0.0281 |
| +DKD [55] | 0.0138 | 0.0389 | **0.0577** | 0.0265 | 0.0325 |
| Student | 0.0094 | 0.0327 | 0.0526 | 0.0210 | 0.0275 |
| +SEvo | 0.0107 | 0.0364 | 0.0576 | 0.0236 | 0.0304 |
| +DKD | **0.0166** | **0.0407** | 0.0568 | **0.0289** | **0.0341** |

Specifically, the distance between each pair $v_i, v_j$ is estimated using a cosine similarity distance function:

$$d_{ij} = 2 - 2 \frac{\mathbf{e}_i^T \mathbf{e}_j}{\|\mathbf{e}_i\|_2 \|\mathbf{e}_j\|_2}. \tag{41}$$

Then, $K$-nearest neighbors are selected for each node; that is

$$(i, j) \in \mathcal{E}, \ i \neq j \text{ iff } |\{k \neq i : d_{ik} \leq d_{ij}\}| \leq K. \tag{42}$$

This sparsification is neccessary for several reasons: 1) SEvo with a dense adjacency matrix is computationally prohibitive to conduct; 2) Generally, only the top-ranked neighbors are reliable for next distillation. Finally, the adjacency matrix is obtained through reweighting and symmetrizing:

$$w_{ij} = \hat{w}_{ij} + \hat{w}_{ji}, \quad \hat{w}_{ij} := \exp(-d_{ij}/\tau),$$

where $\tau > 0$ is the kernel bandwidth parameter.

Table 5 reports the results of the SASRec backbone with different embedding sizes (200 versus 20). Although a student equipped with SEvo can only derive guidance from the teacher in terms of embedding modeling, it has surpassed RKD and HTD that focus on feature/output alignments. Recall that SEvo only needs to access the teacher model once for adjacency matrix construction, whereas other knowledge distillation approaches require accessing the teacher model for each update. SEvo is arguably an efficient tool for transferring embedding knowledge. Nevertheless, SEvo alone cannot be expected to facilitate the learning of the other modules, which consequently is still inferior to state-of-the-art methods such as DKD. Fortunately, SEvo and DKD can work together to further boost the recommendation performance.

## D Additional Experimental Results

### D.1 SEvo for GNN-based models

Table 6: Beauty recommendation performance comparison. SEvo-enhanced AdamW is applied to LESSR and MAERec.

| | HR@1 | HR@5 | HR@10 | NDCG@5 | NDCG@10 |
|---|---|---|---|---|---|
| LESSR | 0.0088 | 0.0322 | 0.0506 | 0.0205 | 0.0264 |
| +SEvo | 0.0126 | 0.0405 | 0.0625 | 0.0267 | 0.0338 |
| Improv. | 43.2% | 26.0% | 23.5% | 30.4% | 27.9% |
| MAERec | 0.0113 | 0.0424 | 0.0662 | 0.0269 | 0.0346 |
| +SEvo | 0.0120 | 0.0441 | 0.0677 | 0.0283 | 0.0358 |
| Improv. | 6.7% | 4.0% | 2.3% | 4.9% | 3.6% |

In Section 3 we have comprehensively validated the effectiveness of SEvo for classic sequence models. It is also of interest to explore the impact on GNN-based models that have learned certain

structural information. Table 6 reports the results on LESSR and MAERec: SEvo not only facilitates the learning of LESSR, but also helps MAERec that already utilizes global graph information. This implies that previous efforts fail to fully exploit structural information, while SEvo demonstrates superior performance in this regard.

## D.2   $\mathcal{J}_{smoothness}$ as a Regularization Term

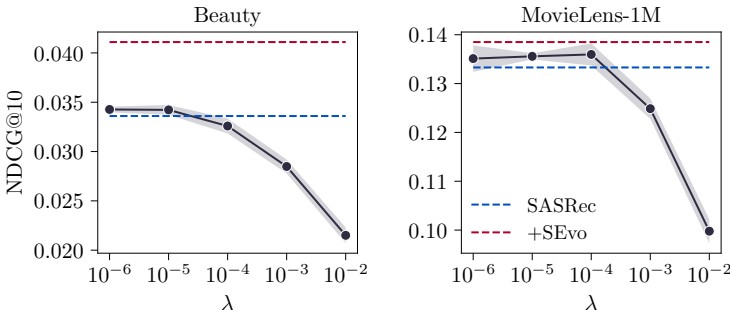

Figure 7: Smoothness constraints through a regularization term.

Structural information may be injected by imposing $\mathcal{J}_{smoothness}$ as a regularization term; that is,

$$\min_{\mathbf{E},\boldsymbol{\theta}} \quad \mathcal{L}(\mathbf{E},\boldsymbol{\theta}) + \lambda\mathcal{J}_{smoothness}(\mathbf{E};\mathcal{G}), \tag{43}$$

where $\lambda \geq 0$ is a hyperparameter governing the degree of smoothness. We conduct this ablation study in Figure 7 with a $\lambda$ from $10^{-6}$ to $0.01$. As can be seen, incorporating a smoothness regularization term could slightly improve the recommendation performance, but it is not optimal. SEvo performs better because the gradient of the regularization term may be in conflict with the primary loss function.

## D.3   $L$-layer Approximation

Table 7: SEvo using different approximation layers $L$.

|  | Beauty | | | MovieLens-1M | | |
|---|---|---|---|---|---|---|
|  | HR@1 | HR@10 | NDCG@10 | HR@1 | HR@10 | NDCG@10 |
| $L$=0 | 0.0124 | 0.0664 | 0.0353 | 0.0465 | 0.2487 | 0.1321 |
| $L$=1 | 0.0140 | 0.0717 | 0.0388 | 0.0498 | 0.2562 | 0.1372 |
| $L$=2 | 0.0152 | 0.0740 | 0.0403 | 0.0511 | **0.2589** | **0.1389** |
| $L$=3 | **0.0154** | **0.0759** | **0.0411** | **0.0517** | 0.2567 | 0.1385 |
| $L$=4 | 0.0153 | 0.0755 | 0.0408 | 0.0510 | 0.2576 | 0.1383 |
| $L$=5 | 0.0150 | 0.0750 | 0.0403 | 0.0492 | 0.2581 | 0.1382 |

As $L$ increases, SEvo gets closer to the exact solution while accessing higher-order neighborhood information. Table 7 lists the performance of different layers, which reaches its peak around $L = 3$ and starts to decrease then. A possible reason is that the higher-order information is over-smoothed and thus not as reliable and easy to use as the lower order information. Similar phenomena have been found in previous works [50, 8] on applying GNNs to recommendation.

## D.4   Training and Inference Times

The time complexity of SEvo is mainly determined by the arithmetic operations of $\tilde{\mathbf{A}}^l \Delta\mathbf{E}, l = 1, 2, \dots, L$. Assuming that the number of non-zero entries of $\tilde{\mathbf{A}}$ is $S$, the complexity required is about $\mathcal{O}(LSd)$. Because the recommendation datasets are known for high sparsity (*i.e.*, $S$ is very small), the actual computational overhead can be reduced to a very low level, almost negligible. Table 8 provides the actual training and inference times.

Table 8: Training and inference times. The wall time (seconds) here is evaluated on an Intel Xeon E5-2620 v4 platform and a single GTX 1080Ti GPU, while the results in Table 3 are tested on an Intel Xeon CPU E5-2680 v4 platform and a single RTX 3090 GPU.

| | | Beauty | | | Toys | | | Tools | | | MovieLens-1M | | |
|---|---|---|---|---|---|---|---|---|---|---|---|---|---|
| | | Training | Inference | Epochs | Training | Inference | Epochs | Training | Inference | Epochs | Training | Inference | Epochs |
| GNN | LightGCN | 2000.50 | 1.07 | 1000 | 1461.14 | 0.97 | 900 | 922.60 | 0.78 | 600 | 6898.42 | 1.95 | 600 |
| | SR-GNN | 25837.60 | 14.52 | 300 | 13711.61 | 11.82 | 200 | 9455.18 | 12.23 | 150 | 126129.93 | 6.16 | 150 |
| | LESSR | 19686.60 | 14.35 | 300 | 15923.59 | 9.67 | 300 | 10994.38 | 8.30 | 300 | 203226.08 | 5.02 | 300 |
| | MAERec | 23956.43 | 3.60 | 100 | 43233.90 | 2.60 | 200 | 16920.16 | 2.31 | 100 | 42586.58 | 2.41 | 200 |
| MF or RNN/Transformer | MF-BPR | 1781.65 | 0.96 | 1000 | 1508.73 | 0.91 | 1000 | 1326.93 | 0.71 | 1000 | 6837.01 | 1.95 | 600 |
| | +SEvo | 1937.32 | 0.96 | 1000 | 1572.91 | 0.91 | 1000 | 1510.23 | 0.71 | 1000 | 7128.00 | 1.95 | 600 |
| | GRU4Rec | 927.98 | 1.98 | 300 | 646.51 | 1.77 | 300 | 638.57 | 1.65 | 300 | 12116.04 | 1.78 | 300 |
| | +SEvo | 987.47 | 1.98 | 300 | 791.19 | 1.77 | 300 | 661.83 | 1.65 | 300 | 12387.25 | 1.78 | 300 |
| | SASRec | 445.68 | 2.16 | 200 | 413.35 | 2.30 | 200 | 353.10 | 1.81 | 200 | 919.27 | 1.24 | 200 |
| | +SEvo | 469.27 | 2.16 | 200 | 480.37 | 2.30 | 200 | 378.25 | 1.81 | 200 | 954.77 | 1.24 | 200 |
| | BERT4Rec | 1470.76 | 2.03 | 500 | 1330.58 | 1.71 | 500 | 1092.10 | 1.51 | 500 | 1131.62 | 1.18 | 500 |
| | +SEvo | 1965.97 | 2.03 | 500 | 1595.66 | 1.71 | 500 | 1374.72 | 1.51 | 500 | 1243.08 | 1.18 | 500 |
| | STOSA | 2253.98 | 9.84 | 500 | 2049.00 | 8.42 | 500 | 1827.60 | 6.65 | 500 | 2220.18 | 1.98 | 500 |
| | +SEvo | 2491.54 | 9.84 | 500 | 2231.11 | 8.42 | 500 | 1879.62 | 6.65 | 500 | 2259.66 | 1.98 | 500 |

