# OpenReview forum: "Graph-enhanced Optimizers for Structure-aware Recommendation Embedding Evolution"
_NeurIPS.cc/2024/Conference — NeurIPS 2024 poster_

### Official Review · Reviewer_9GVe · 2024-07-07

**Soundness:** 3
**Presentation:** 4
**Contribution:** 3
**Rating:** 6
**Confidence:** 4

**Summary:**

In this paper, the authors propose a novel optimization algorithm that is talored for recommender systems. It incorporates graph structural information into the optimization process, aleviating the burden of performing GNN for RS. The convergence of the algorithm is theoretically demonstrated. Besides, it could be incorporated into existing well-performed optimizers like AdamW. The experiments are conducted to test its effectiveness on different types of recommendation models and consistent performance improvements are observed.

**Strengths:**

1. This paper is innovative in its consideration of utilizing graph information the algorithm level, not model level, for recommender systems. The proposed new algorithm is theoretically guaranteed to converge.
2. The application of the proposed algorithm is widely discussed, including its incorporation into existing popular optimizers and its combination with knowledge distillation for recommender systems.
3. The experiments are extensively conducted, including different size of datasets, different types of recommendation backbones and baselines. The results demonstrate both the effectiveness and efficiency of the proposed algorithm for improving existing recommendation models.

**Weaknesses:**

1. The technique in Section 2.4 is a little too specific for AdamW.
2. Table 5 seems to have an error. In the last row, 0.568 is not the best result. Is it just a typo? Or does +DKD not further improve the performance in this case?

**Questions:**

In Table 6, the proposed optimization algorithm even improves the GNN-based recommendation models. What is the reason for this phenomenon?

**Limitations:**

The authors have addressed the limitations in Section 5 and Appendix.

---

> ### Author Rebuttal · Authors · 2024-08-03
>
> We thank the reviewer for finding our work innovative.
> Each comment (presented in *italics*) is followed by its corresponding response.
>
> > *W1: The technique in Section 2.4 is a little too specific for AdamW.*
>
> Thank you for your feedback.
> AdamW was chosen as an example because it is the most widely used optimizer in recommender systems other than Adam.
> Similar modifications can also be implemented for other optimizers if they incorporate an **exponential moving average momentum term** (e.g., Adamax [R1], AMSGrad [R2], AdaBelif [R4]) and can decouple the weight decay regularization [R3].
>
> > *W2: Table 5 seems to have an error. In the last row, 0.568 is not the best result. Is it just a typo? Or does +DKD not further improve the performance in this case?*
>
> |Seed| HR@1 | HR@5| HR@10| NDCG@5| NDCG@10|
> |:-:|:-:|:-:|:-:|:-:|:-:|
> |0	|0.01744	|0.04109	|0.05755	|0.02956	|0.03489|
> |1	|0.01646	|0.04087	|0.05643	|0.02907	|0.03404|
> |2	|0.01655	|0.04092	|0.05666	|0.02900	|0.03406|
> |3	|0.01641	|0.04092	|0.05612	|0.02878	|0.03370|
> |4	|0.01619	|0.03971	|0.05746	|0.02811	|0.03381|
>
> Thank you for pointing out this mistake.
> **We reviewed the results (for the 5 seeds shown above) and found that '+DKD' indeed does not improve the HR@10 performance.**
> Furthermore, the overall comparison in Table 2 indicates the common conclusion: the application of SEvo is primarily beneficial for the top-ranked targets.
> This is understandable as SEvo encourages the related nodes to be closer; conversely, it may adversely affect the targets being less related to the historical items.
> To address this problem, graphs describing multiplex relations should be introduced, which is left as future work as discussed in Section 5.
> We will further analyze this interesting observation in the revised manuscript.
> Thank you once again for your meticulous review.
>
>
> > *Q1: In Table 6, the proposed optimization algorithm even improves the GNN-based recommendation models. What is the reason for this phenomenon?*
>
>
> We believe that it is challenging to exploit both structural and sequential information for these GNN-based recommenders. LESSR and MAERec have developed sophisticated architectures for this purpose; however, they remain suboptimal in effectively utilizing either structural or sequential information alone. To be specific, the local session graph used by LESSR is insufficient to model the necessary structural and sequential information. In contrast, MAERec has improved its performance in this regard due to the use of two separate modules; however, effectively fusing these modules remains a challenging task.
>
>
> [R1] Kingma D. P., et al. Adam: A method for stochastic optimization. ICLR, 2015.
>
> [R2] Reddi S. J., et al. On the convergence of adam and beyond. ICLR, 2018.
>
> [R3] Loshchilov I., et al. Decoupled weight decay regularization. ICLR, 2019.
>
> [R4] Adabelief optimizer: Adapting stepsizes by the belief in observed gradients. NeurIPS, 2020.

---

### Official Review · Reviewer_feMV · 2024-07-11

**Soundness:** 3
**Presentation:** 3
**Contribution:** 3
**Rating:** 6
**Confidence:** 4

**Summary:**

This paper proposes Structure-aware Embedding Evolution (SEvo) to improve recommender systems by directly integrating graph structural information into embeddings. Unlike traditional methods, authors propose guide embedding update momentum with graph smoothing regularization. The proposed method can be integrated with a wide range of optimizers for neural networks, e.g., AdamW. The proposed method significantly increases performance metrics on several recommender datasets for standard models and outperforms GNNs.

**Strengths:**

* Novel intriguing view on the problem of preserving structural information for recommender systems
* Plug-n-play design of the method so that it can be easily transferred to any recommendation architecture
* Solid performance gains
* Faster than training GNNs

**Weaknesses:**

* The paper is difficult to follow. It would be nice to see a simple summary of the SEvo pipeline in the form of a scheme or algorithm.
* The SEvo formulae use the normalized adjacency matrix, so propagation of the gradients over the sampled node neighborhood is required. Suppose we have a large graph with a relatively high degree of each node. Mini-batch may have poorly correlated nodes, so many node embeddings should be updated simultaneously. This can lead to memory consumption issues and a notable increase in training time.
* The tables report only the average time across different datasets. The detailed computational (or non-aggregated graphs with time)  and space complexity is required to understand the ability of the method to scale
* SEvo accelerates momentum only over first-order neighbors. However, for some graph-related tasks, it is critical to handle long-range dependencies.

**Questions:**

* How does SEvo work in mini-batch fashion? Does it require specific batch preparations, or should it be smaller on average?
* How does the model training time scale with the size of the graph? Could you provide a graph epoch time vs. graph size e.g. for a SASRec?
* Can high-order proximity / long-range dependencies be incorporated using SEvo?

**Limitations:**

* The time and space complexity analyses of the method are required to understand its scalability.

---

> ### Author Rebuttal · Authors · 2024-08-03
>
> We thank the reviewer for finding our work intriguing.
> Each comment (presented in *italics*) is followed by its corresponding response.
>
> > *W1: The paper is difficult to follow. It would be nice to see a simple summary of the SEvo pipeline in the form of a scheme or algorithm.*
>
> Thank you for your constructive feedback.
> The SEvo pipeline can be summarized as follows (the algorithms for SEvo-enhanced SGD/Adam/AdamW are detailed in Appendix B.1):
> 1. Compute gradients for embeddings;
> 2. Update moment estimates (with certain modifications discussed in Section 2.4);
> 3. Smooth the variations according to Eq. (7);
> 4. Update the embeddings.
>
> > *W2: The SEvo formulae use the normalized adjacency matrix, so propagation of the gradients over the sampled node neighborhood is required. Suppose we have a large graph with a relatively high degree of each node. Mini-batch may have poorly correlated nodes, so many node embeddings should be updated simultaneously. This can lead to memory consumption issues and a notable increase in training time.*
>
> As discussed in Appendix D.4, the complexity required for SEvo is comparable to that of the simplest GCN solely with neighborhood aggregation. Therefore, SEvo can be readily used for large graphs to which other GCNs can be applied.
>
> > *W3/Q2: The tables report only the average time across different datasets. The detailed computational (or non-aggregated graphs with time) and space complexity is required to understand the ability of the method to scale. How does the model training time scale with the size of the graph? Could you provide a graph epoch time vs. graph size e.g. for a SASRec?*
>
> We present the computational and memory costs below (more training and inference times have been detailed in Appendix D.4). The graph size increases from Tools to Clothing.
>
> | | Tools | Beauty | Electronics | Clothing |
> |:-: |:-: |:-: |:-: |:-: |
> | #Users | 16,638 | 22,363 | 728,489 | 1,219,337 |
> | #Items | 10,217 | 12,101 | 159,729 | 376,378 |
> | #Edges | 134,476 | 198,502 | 6,737,580 | 11,282,445 |
>
> | Method (second/epoch) | Tools | Beauty | Electronics | Clothing |
> |:-: |:-: |:-: |:-: |:-: |
> | SR-GNN | 63.03s | 86.13s | 1,653.53s | 2,909.38s |
> | LESSR | 36.65s | 65.62s | 1,846.30s | 3,615.53s |
> | MAERec | 169.20s | 239.56s | 15,464.64s | 14,017.92s |
> | SASRec | 1.76s | 2.23s | 19.94s | 25.20s |
> | SASRec+SEvo | 1.89s | 2.35s | 22.16s | 33.47s |
>
> | Method (GPU memory) | Tools | Beauty | Electronics | Clothing |
> |:-: |:-: |:-: |:-: |:-: |
> | SR-GNN | 1,214M | 1,212M | 2,056M | 2,328M |
> | LESSR | 1,618M | 1,660M | 19,868M | 20,772M |
> | MAERec | 1,952M | 1,972M | 4,664M | 7,478M |
> | SASRec | 2,064M | 2,036M | 2,510M | 3,282M |
> | SASRec+SEvo | 2,074M | 2,046M  | 2,908M | 4,080M |
>
>
> We have the following observations:
> 1) Compared to vanilla SASRec, the implementation of SEvo incurs minimal computational and memory costs.
> 2) Due to the high sparsity of recommendation datasets, the additional cost increases acceptably as the graph size increases.
> 3) The costs associated with SEvo are negligible compared to other GNN-based sequence models, including SR-GNN, LESSR, and MAERec. Surprisingly, the training time for these models on Tools (the smallest dataset) greatly exceeds the time required for SEvo on Clothing (the largest dataset). This limitation hinders their application in real recommendation scenarios.
>
> > *W4/Q3: SEvo accelerates momentum only over first-order neighbors. However, for some graph-related tasks, it is critical to handle long-range dependencies. Can high-order proximity / long-range dependencies be incorporated using SEvo?*
>
> Thanks for the comment but there is a misunderstanding about this.
> SEvo can intrinsically handle long-range dependencies as reflected in Eq. (7).
> Moreover, as investigated in Appendix D.3 and Table 7, SEvo can benefit from long-range dependencies when $L > 1$, with better results when $L=$ 2 or 3.
>
>
> > *Q1: How does SEvo work in mini-batch fashion?*
>
> **Graph sampling** methods [R1, R2] can be applied in a manner similar to that employed for other GCNs. Furthermore, it is worth noting that **graph partition** approaches [R3, R4] are more appropriate here, as the items/entities in a mini-batch are not necessarily included in the graph utilized for SEvo. When the original graph is too large for practical applications, it can be **pre-sliced** into multiple subgraphs for subsequent training. Moreover, if these smaller subgraphs are mutually exclusive, parallel updates can be performed for better acceleration and accuracy.
>
> > *Q1.5:  Does it require specific batch preparations, or should it be smaller on average?*
>
> Do you mean the sampled/sliced sub-graphs should be smaller on average?
> We think the bigger the better, if possible.
>
>
> > *L1: The time and space complexity analyses of the method are required to understand its scalability.*
>
> As discussed in Appendix D.4, the **time complexity** of SEvo is mainly determined by the arithmetic operations of $\mathbf{\tilde{A}}^l \Delta \mathbf{E}, l=1,2, \ldots, L$.
> Assuming that the number of non-zero entries of $\mathbf{\tilde{A}}$ is $S$, the complexity required is about $\mathcal{O}(LSd)$. On the other hand, the **additional space complexity** of SEvo is $\mathcal{O}(S)$ for the storage of the normalized adjacency matrix.
> Because the recommendation datasets are known for high sparsity (i.e., $S$ is very small), the actual overhead can be reduced to a very low level.
>
>
> [R1] Hamilton W. L., et al. Inductive representation learning on large graphs. NeurIPS, 2017.
>
> [R2] Zou D., et al. Layer-dependent importance sampling for training deep and large graph convolutional networks. NeurIPS, 2019.
>
> [R3] Chiang W., et al. Cluster-GCN: An efficient algorithm for training deep and large graph convolutional networks. KDD, 2019.
>
> [R4] Liu X., et al. Survey on graph neural network acceleration: An algorithmic perspective. IJCAI, 2022.

---

> > ### Comment · Reviewer_feMV · 2024-08-09
> > **Acknowledgment of Clarifications and Updated Final Rating**
> >
> > Thank you for your detailed and thoughtful rebuttal. I appreciate the time and effort you have taken to address my concerns.
> >
> > Having carefully considered your responses, I am satisfied with the clarifications and additional insights provided. Your explanations have resolved the issues I initially raised, and I now have a clearer understanding of the contributions and significance of your work.
> >
> > I appreciate your efforts and will be revising my final rating to reflect the improvements made.

---

> > > ### Author Response · Authors · 2024-08-10
> > >
> > > We are delighted to learn that our responses have addressed your concerns. We sincerely appreciate the time you spent reviewing our paper!

---

### Official Review · Reviewer_DLFs · 2024-07-18

**Soundness:** 3
**Presentation:** 2
**Contribution:** 3
**Rating:** 5
**Confidence:** 3

**Summary:**

The paper introduces Structure-aware Embedding Evolution (SEvo), a novel embedding update mechanism for recommender systems. SEvo directly integrates graph structural information into embeddings, ensuring that related nodes evolve similarly with minimal computational overhead. This approach differs from traditional Graph Neural Networks (GNNs), which typically serve as intermediate modules. SEvo is designed to enhance existing optimizers, particularly AdamW, to improve recommendation performance by incorporating moment estimate corrections. Theoretical analysis confirms the convergence properties of SEvo, and experiments demonstrate consistent improvements across various models and datasets.

**Strengths:**

1. The paper proposes a new method to enhance over smoothing during the backward pass.

2. The method can be naturally integrated with momentum-based optimizers

**Weaknesses:**

1. Uncleared relationship related to recommender system

2. Key points need further explanation.

**Questions:**

1. I do not see the relationship between your method and recommendation task. Your method aims to change gradient direction based on graph topology. It is more suitable to study it under more general graph datasets. Recommendation task has no relationship with your method.

2. In line 116, the author mentioned “These two criteria inherently conflict to some extent”. Why are structure-aware and direction-aware inherently in conflict? Do you have any explanation on this point?

3. We have already enhanced smoothness during the forward-pass during neighborhood aggregation. Why do authors think it is necessary to further enhance it during the backward pass? Will it lead to more severe over smoothing problem?

**Limitations:**

See weakness and questions

---

> ### Author Rebuttal · Authors · 2024-08-03
>
> We thank the reviewer for finding our work novel.
> Each comment (presented in *italics*) is followed by its corresponding response.
>
> > *W1/Q1: Uncleared relationship related to recommender system. It is more suitable to study it under more general graph datasets. Recommendation task has no relationship with your method.*
>
> Thank you for pointing out the lack of a clear statement of the motivation in this manuscript.
> We agree that **technically** SEvo can be applied to a wider range of graph datasets; however, it is more appropriate to discuss its application in a recommendation scenario for two practical considerations.
>
> 1. Embedding is particularly important in modern recommender systems and its quality directly affects the subsequent decisions. However, due to **data sparsity** [R2], **millions of item embeddings** cannot be consistently updated through a simple recommendation-driven loss function. To this end, SEvo-enhanced AdamW introduces a graph regularization framework for consistent embedding evolution while simultaneously modifying AdamW to effectively address the challenges posed by extremely sparse gradients.
>
> 2. Compared to embedding learning for general graph datasets [R1], a major challenge for recommendation is **how to effectively injecting structural information while leveraging other types of information** (e.g., sequential information). SEvo excels in this aspect as it has minimal impact on the forward process (this can be empirically demonstrated through comparisons with other GNN-based sequence recommenders).
>
> In summary, SEvo-enhanced AdamW is specifically designed to address the challenges of data sparsity and injecting multiple types of information, which rarely co-exist in general graph datasets. We will further emphasize these challenges in the revised manuscript.
>
> > *Q2: Why are structure-aware and direction-aware inherently in conflict? Do you have any explanation on this point?*
>
> Given an adjacency matrix $\mathbf{\tilde{A}}$, the smoothest direction is along its principal eigenvector $\mathbf{D}^{1/2} \mathbf{1}$, ensuring that $\mathcal{J}_{smoothness}(\mathbf{D}^{1/2}) = 0$. However, the region around this smoothest direction tends to be an infeasible descent direction. Therefore, we have to resort to Eq. (6) for a trade-off.
>
> > *Q3: We have already enhanced smoothness during the forward-pass during neighborhood aggregation. Why do authors think it is necessary to further enhance it during the backward pass? Will it lead to more severe over smoothing problem?*
>
> Thank you for your insightful comment.
> We agree that if the forward process already involves neighborhood aggregation, re-enhancing smoothness during the backward pass may exacerbate the over-smoothing issue. However, as stated in the introduction, SEvo is not intended to replace those sophisticated GNNs but rather to offer an easy-to-use and plug-and-play alternative for structural information learning.
> In addition, if the base model fails to exploit the structural information **adequately** during the forward pass (e.g., LESSR and MAERec), SEvo can still facilitate the learning (the comparison on Beauty as an example):
>
> | | HR@5 | HR@10 | NDCG@5 | NDCG@10 |
> |:-: |:-: |:-: |:-: |:-: |
> | LESSR | 0.0322 | 0.0506 | 0.0205 | 0.0264 |
> | +SEvo | **0.0405** | **0.0625** | **0.0267** | **0.0338** |
> | Improv. | 26.0% | 23.5% | 30.4%  | 27.9%  |
> | MAERec| 0.0424 | 0.0662 | 0.0269 | 0.0346 |
> | +SEvo | **0.0441** | **0.0677** | **0.0283** | **0.0358** |
> | Improv. | 4.0% | 2.3% | 4.9% | 3.6% |
>
> [R1] Chami I., et al. Low-dimensional hyperbolic knowledge graph embedding. arXiv preprint, 2020.
>
> [R2] Chen Z., et al. A systematic literature review of sparsity issues in recommender systems. TORS, 2024.

---

> > ### Comment · Reviewer_DLFs · 2024-08-12
> > **Keep score unchanged**
> >
> > I have acknowledged the rebuttal from authors and keep my score unchanged.

---

> > > ### Author Response · Authors · 2024-08-12
> > >
> > > We sincerely thank you for your time and valuable comments. If you have any further questions please let us know.

---

### Official Review · Reviewer_Um97 · 2024-07-30

**Soundness:** 3
**Presentation:** 4
**Contribution:** 3
**Rating:** 6
**Confidence:** 3

**Summary:**

This paper proposes SEvo, an embedding updating mechanism that directly injects the graph information into the optimization process. This paper points out two critical criteria for directly injecting graph structure information into the embedding updating process for recommendation. Based on the proposed two criteria, this paper makes efforts to derive a solution named SEvo for injecting the graph structure information directly. SEvo is model-agnostic and can be implemented in various optimizers. The experiments are detailed, and the algorithm is theoretically guaranteed.

**Strengths:**

1. This paper is well-motivated and well-organized, making this paper easy to understand. This paper first proposes two criteria and derives the final form of SEvo. I appreciate the efforts of the authors to make this process so clear.
2. The experiments are detailed. Experiments on various datasets showcase the effectiveness of SEvo, with detailed ablation studies.
3. The proposed method is easy to implement. SEvo can be integrated into various optimizers without complex modification.

**Weaknesses:**

The reason behind the success of SEvo on large-scale datasets remains unclear. I am extremely curious about this. The improvement is unbelievably huge, making me doubt the reported results. A level of 5% in practice industrial application is extremely huge. However, the experiment results show that SASRec equipped with SEvo performs twice as well as the vanilla SASRec. I believe that if the reported results are true, this performance even exceeds the SOTA method by a large margin since SASRec is still a strong baseline in practice. Has the author carefully tuned the base model?

**Questions:**

I am curious about the differences in gradient descent between SEvo and methods that explicitly model the neighborhood relationship with graph structure (e.g., LightGCN). The modified embedding updating mechanism in SEvo also includes components of the adjacent matrix, which looks similar to the graph propagation mechanism in GNN-based methods. Could you please give an example showcasing the differences in gradient descent (e.g., comparing the gradient descent processes of LightGCN and MF-BPR+SEvo)?

**Limitations:**

See weaknesses and questions.

---

> ### Author Rebuttal · Authors · 2024-08-03
>
> We thank the reviewer for finding our work well-motivated and well-organized.
> Each comment (presented in *italics*) is followed by its corresponding response.
>
> > *W1: The reason behind the success of SEvo on large-scale datasets remains unclear. I am extremely curious about this. The improvement is unbelievably huge, making me doubt the reported results. A level of 5% in practice industrial application is extremely huge. However, the experiment results show that SASRec equipped with SEvo performs twice as well as the vanilla SASRec. I believe that if the reported results are true, this performance even exceeds the SOTA method by a large margin since SASRec is still a strong baseline in practice. Has the author carefully tuned the base model?*
>
> Thank you for your constructive feedback.
> We did carefully tune the base model and the following hyper-parameters were **grid-searched**:
>
> | Parameter | Range |
> |:-: |:-: |
> |Learning rate | \{1e-4, 5e-4, 1e-3, 5e-3\} |
> | Weight decay | [0, 1e-4] |
> | Dropout rate | [0, 0.4] |
> | Batch size | \{1024, 2048, 4096\} |
>
> The **best checkpoint**, determined by the validation metric, was used for the final comparison.
> **In the attached PDF within the global response, we illustrate partial results in terms of the Electronics dataset.**
> We were as surprised as the reviewers by SEvo's success with these large-scale datasets.
> At this point, the following observations may provide valuable insights.
>
> - The training process of the base model exhibits increased instability on larger-scale datasets. This can be attributed to the sampling randomness: only a small fraction of items are sampled for training within a mini-batch, resulting in the remaining embeddings receiving zero gradients and being updated along the outdated and inconsistent directions. This problem gets worse as the dataset grows larger and sparser. SEvo and the specific modifications to AdamW (Section 2.4) have a positive impact on this issue. In this scenario, even the embeddings of highly inactive items will be updated appropriately throughout the training process.
>
> - In real industrial applications, this performance gap may not be as large because the embeddings are typically trained more adequately beyond a link prediction task [R2]. Hence, the data sparsity problem can be greatly alleviated. Moreover, the rich side information (e.g., attributes [R1], behaviors [R3]) can further minimize the impact of poor embedding quality.
>
> > *Q1: Could you please give an example showcasing the differences in gradient descent (e.g., comparing the gradient descent processes of LightGCN and MF-BPR+SEvo)?*
>
> Thank you for your insightful question.
> We investigate the gradient descent process of LightGCN and find an interesting connection to SEvo.
> For a $L$-layer LightGCN, it can be formulated as follows
> $$
> \mathbf{F} = \psi (\mathbf{E}) := \sum_{l=0}^L \alpha_l \mathbf{\tilde{A}}^l \mathbf{E},
> $$
> where $\alpha_l (l=0, \ldots, L)$ represent the layer weights.
> According to the linear nature of the gradient operator, it can be obtained that
> $$
> \nabla_{\mathbf{E}} \mathcal{L} = \psi (\nabla_{\mathbf{F}} \mathcal{L}).
> $$
> Hence, denoted by $\zeta(\cdot)$ the gradient processing procedure of an optimizer,
> we can establish that LightGCN is identical to the following system ($\mathbf{F}(t) := \mathbf{F}_t$ due to OpenReview's inability to recognize '\_' in very long formulas):
>
> $$
> \mathbf{F}(t) = \psi( \mathbf{E}(t) ) = \psi( \mathbf{E}(t-1) - \eta \Delta \mathbf{E}(t-1) ) = \psi( \mathbf{E}(t-1) ) - \eta \psi( \Delta \mathbf{E}(t-1) ) = \mathbf{F}(t-1) - \eta \psi \circ \zeta \circ \psi ( \nabla_{\mathbf{F}} \mathcal{L} ).
> $$
>
> When $\zeta(\cdot)$ is an identity mapping (i.e., standard gradient descent), LightGCN is equivalent to MF-BPR with SEvo being applied twice at each update.
> However, when $\zeta(\cdot)$ is not an identity mapping (e.g., an optimizer with momentum or weight decay is integrated), they cannot be unified into a single system.
> Compared to explicit GNNs,
> SEvo is easy-to-use and has minimal impact on the forward pass, making it more suitable for assisting recommenders in simultaneously utilizing multiple types of information.
> These connections in part justify why SEvo can inject structural information directly. We will emphasize this point in the revised manuscript.
>
> [R1] Chen Q., et al. Behavior sequence transformer for e-commerce recommendation in Alibaba. DLP-KDD, 2019.
>
> [R2] Bai T., et al. A contrastive sharing model for multi-task recommendation. WWW, 2022.
>
> [R3] Yang Y., et al. Multi behavior hypergraph-enhanced transformer for sequential recommendation. KDD, 2022.

---

> > ### Comment · Reviewer_Um97 · 2024-08-09
> > **My concern has been addressed.**
> >
> > Thank the author for addressing my concern. I will keep my rating.

---

> > > ### Author Response · Authors · 2024-08-09
> > >
> > > We thank you for the engagement with our work and for your effort during the rebuttal.

---

### Author Rebuttal · Authors · 2024-08-03

We would like to thank all reviewers for reviewing our paper and providing us with their insightful comments.
We are excited that the reviewers found our work novel and well-written, and pleased that they were satisfied with both our theoretical analysis and experimental results.
We do our best to respond to reviewers' comments so that the work is of a higher standard.
Some of the major critiques are listed below; each reviewer's comments will be addressed point by point.

1. Reviewer Um97 expressed concerns regarding the substantial success of SEvo on large-scale datasets. We have thoroughly examined our reproducibility results and ensured that the comparisons are conducted fairly.
2. Reviewer DLFs thought that SEvo may apply to general graph datasets. We believe that both the motivation and design are specifically tailored to address the two major challenges of recommendation.
3. Reviewer feMV had concerns about the computational and space complexity of SEvo. We have carried out a complexity analysis supported by experimental evidence, which shows that SEvo is scalable and efficient compared to the simplest GCN.
4. Reviewer 9GVe carefully reviewed the paper and found a few typos. We have made modifications in the revised manuscript.

Please check our rebuttal and let us know if you have any further questions.

---

### Decision · Program_Chairs · 2024-09-25

**Decision:**

Accept (poster)

**Comment:**

Post rebuttal, all reviewers have agreed to accept the paper. I concur that the paper offers a valuable graph embedding update mechanism for recommender models. Therefore, I recommend acceptance. Please ensure that the necessary revisions are incorporated into the final version.